# Don't Pay Attention, PLANT It: Pretraining Attention via Learning-to-Rank

## Abstract

State-of-the-art Extreme Multi-Label Text Classification models rely on multi-label attention to focus on key tokens in input text, but learning good attention weights is challenging. We introduce PLANT—**P**retrained and **L**everaged **A**tte**NT**ion—a plug-and-play strategy for initializing attention. PLANT works by *planting* label-specific attention using a pretrained Learning-to-Rank model guided by mutual information gain. This architecture-agnostic approach integrates seamlessly with LLM backbones (e.g., we consider `Mistral-7B`, `LLaMA3-8B`, `DeepSeek-V3`, and `Phi-3`). PLANT outperforms SOTA methods across tasks like ICD coding, legal topic classification, and content recommendation. Gains are especially pronounced in few-shot settings, with substantial improvements on rare labels. Ablation studies confirm that attention initialization is a key driver of these gains. We make our code and trained models available.

## 1 Introduction

Extreme Multi-Label Text Classification (XMTC) entails assigning the most relevant subset of labels to a given instance from a (very) large label set. This setting emerges naturally in domains featuring vast, structured taxonomies such as e-commerce, legal categorization, and healthcare. In such settings, manual labeling is both costly and error-prone. For example, in clinical settings (Table 1), ICD coding—the task of assigning standardized codes for diagnoses and procedures based on clinical notes (Moons et al., 2020; WHO, 2025)—may be viewed as an instance of XMTC.

| | | |
|---|---|---|
| 428.0: *Congestive heart failure* | 202.8: *Other malignant lymphomas* | 770.6: *Transitory tachypnea of newborn* |
| · · · DIAGNOSES: **Acute congestive heart failure**, Diabetes mellitus, Pulmonary edema · · · | · · · 55 year-old female with **non Hodgkin's lymphoma** and C1 esterase inhibitor deficiency · · · | · · · Chest x-ray: **transient tachypnea of the newborn** with respiratory distress · · · |

Table 1: Examples of clinical text with ICD codes (Wang et al., 2024d; Zhang et al., 2025). Blue: code/label; red bold: disease mentions; teal: other relevant clinical findings.

Building XMTC models is challenging due to the high-dimensional label space and heavily skewed label distributions Bhatia et al. (2016). For example, in ICD coding there can be 170000 unique codes (CDC, 2024). Many are rare: In the MIMIC-III dataset Johnson et al. (2016) approximately 5411 out of 8929 codes appear <10 times. The task is further exacerbated by the often lengthy narratives in clinical texts. For example, in the MIMIC-III dataset, discharge summaries frequently contain detailed clinical histories comprising an average of 709.3 tokens, and often exceeding 1500 tokens (Johnson et al., 2016; 2023; Mullenbach et al., 2021; Nguyen et al., 2023). However, only a small fraction of these tokens are informative for assigning relevant ICD codes.

LLMs can be used zero-shot for XMTC tasks, but this poses challenges. For instance, prompts for such tasks tend to include long and flat label lists, resulting in *attention dilution*: The fixed attention budget is spread thin across thousands of tokens, weakening focus on rare tail labels (Peysakhovich & Lerer, 2023; Vandemoortele et al., 2025). This limitation is similarly evident in long-context retrieval tasks (Kamradt, 2023; Hsieh et al., 2024; Liu et al., 2024a), where LLMs struggle to locate relevant items. Task-specific fine-tuning may address such issues by embedding knowledge of the labels directly into model parameters during training, obviating the need for attention over long label

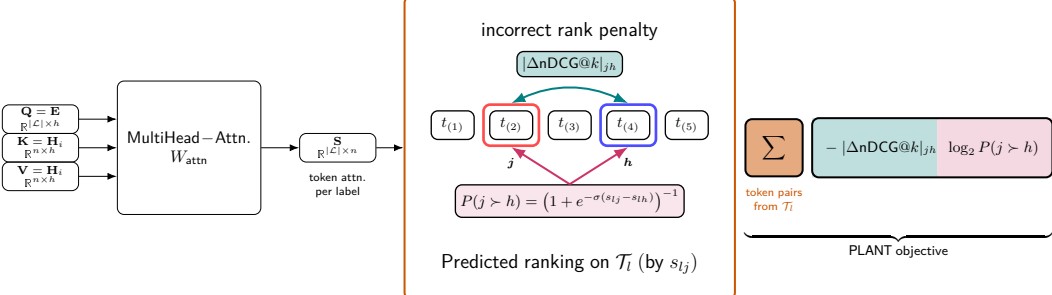

Figure 1: **PLANT** Attention. On the left, the MultiHead−Attention module (Vaswani et al., 2017), parameterized by $\boldsymbol{W}_{\text{attn}}$, takes as input queries $\mathbf{Q} = \mathbf{E}$ (label embeddings), keys $\mathbf{K} = \mathbf{H}_i$, and values $\mathbf{V} = \mathbf{H}_i$, and produces $\mathbf{S} \in \mathbb{R}^{|\mathcal{L}| \times n}$, representing the token-level attention distribution for each label. The **orange box** highlights the set of top-$k$ tokens per label, $\mathcal{T}_l$, selected via Mutual Information Gain $r_{lj}$ between labels and tokens. Within this set, two tokens $j$ (**red**) and $h$ (**blue**) are compared, with $j$ being more relevant than $h$. The MultiHead−Attention module is trained to maximize the probability of correctly ranking tokens $j$ and $h$ ($P(j \succ h)$), while penalizing incorrect rankings in proportion to their impact on the nDCG@k metric if $j$ and $h$ were swapped ($|\Delta\text{nDCG@}k|_{jh}$). Finally, the **summation box** aggregates over all token pairs in $\mathcal{T}_l$, yielding the PLANT objective—(**nDCG term** × **probability term**)—that is optimized to initialize $\boldsymbol{W}_{\text{attn}}$.

lists in the prompt and thus mitigating attention dilution (Yang et al., 2023a; Boukhers et al., 2024; Zhang et al., 2025; Barreiros et al., 2025).

In current approaches to XMTC, *attention mechanisms* Bahdanau et al. (2014) help address the challenges of high-dimensional, skewed label spaces. Existing XMTC models (Lu et al., 2023; Li et al., 2023; Nguyen et al., 2023; Yang et al., 2023b; Chen et al., 2023a; Zhang & Wang, 2024; Luo et al., 2024) almost always include a multi-label attention layer that allocates per-label attention weights to the input tokens (Wang et al., 2023a; Xiong et al., 2023; Yuan et al., 2024; Liu et al., 2025b). Intuitively, this is akin to a dedicated "spotlight" for each label: in high-dimensional spaces, it avoids the inefficiency of a single global focus by creating tailored text representations that highlight most relevant tokens per label. For skewed distributions, this ensures subtle cues for tail labels are not overshadowed by head labels, enabling better prediction of sparse classes.

Regardless of the specific encoder architecture, removing this attention layer significantly harms performance. A recent study by Xiong et al. (2023) highlights the importance of label-specific attention for *product-to-tag matching* by showing that removing this component leads to a sharp drop in P@1 (-15.69 points). Elsewhere, results on *scientific paper classification* show that stacking attention layers further boosts performance: Micro-F1 improves by a few points, showing that deeper attention enhances the model's capacity to represent label-specific features Liu et al. (2025b).

**The premise of this work is that we can be smarter about how we initialize attention module weights**. SOTA XMTC models begin with random label attention weights, requiring ranking all tokens for each label from scratch. This is data-intensive due to the high-dimensional label space. Skewed label distributions exacerbate this issue, as rare labels require even more data. Insufficient data, however, causes models to require more training epochs, often leading to overfitting rather than meaningful generalization—ultimately hurting rare label performance. Studies like Edin et al. (2023) show that SOTA models struggle to predict rare ICD diagnosis codes (Figure 2, left). Models perform similarly across codes with comparable frequencies, indicating that the high proportion of rare codes impacts performance. Correlations between code frequency and F1 score are moderately high, showing that rare codes are predicted less accurately than common ones. This underscores the need for efficient attention mechanisms, as starting with random weights may be suboptimal.

Building on evidence that label-specific attention is pivotal in XMTC—its removal leads to sharp performance drops—we argue that how this attention is initialized is also crucial. To establish the causal link—"poor rare-label performance ← failure to discover shared attention structure ← random initialization of the label-attention layer"—and, at the same time, disentangle initialization effects from downstream training dynamics, we start with a qualitative, diagnostic experiment.

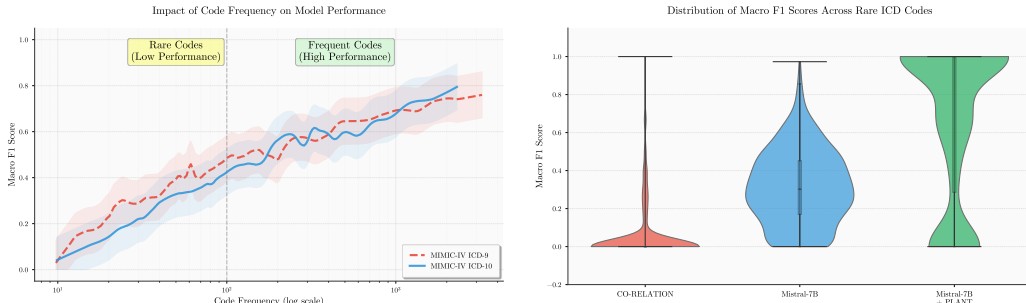

Figure 2: *(Left)* Rare codes have near-zero macro-F1. *(Right)* Macro-F1 distribution on `MIMIC-III-few` for rare codes across CO-RELATION (Luo et al., 2024) (mean=0.054), `Mistral-7B` (0.309), and `Mistral-7B` +PLANT (0.663). `Mistral-7B` +PLANT yields far more rare codes with higher F1. See Section 3 (RQ4).

We use ICD codes, an important illustrative instance of XMTC, as a motivating example. Because ICD codes are hierarchical, codes within the same clinical category are semantically related and should, in principle, induce similar attention patterns over the input note. To test whether learned label attention vectors $\mathbf{S}_l$ reflect this structure, we selected two groups of 50 ICD-10 codes: one common group (respiratory tuberculosis, A15–A19) and one rare group (various rare bacterial infections, A30–A49). Under standard random initialization of the label attention layer, codes in the **rare** group show widely dispersed pairwise cosine similarities (mean 0.75; orange distribution in Figure 3, Left), indicating that the model fails to recover their shared structure. In contrast, the **common** group already shows strong intra-group consistency (mean > 0.98; blue). This stark asymmetry—common codes converge to coherent representations while rare yet semantically similar codes do not—reveals a key failure mode of random-initialized attention on long-tail labels. This motivated PLANT. By seeding the attention layer with mutual-information signals and Learning-to-Rank activations, PLANT boosts intra-group consistency for the rare category to 0.985 (sharp brick-red spike in Figure 3, Left), bringing rare-label representations up to the quality enjoyed by frequent codes.

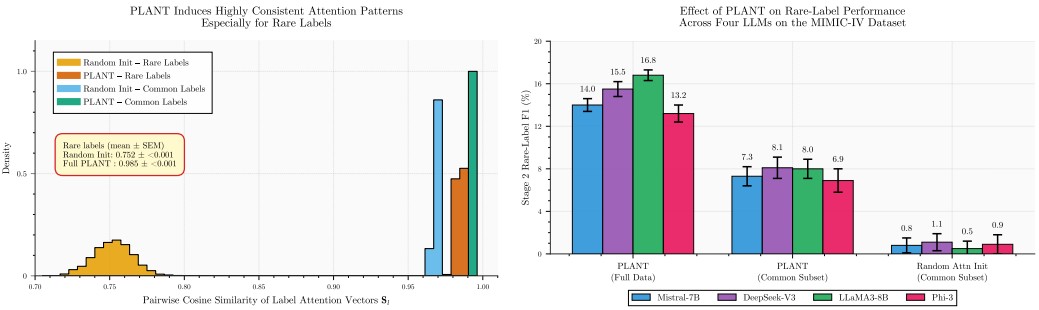

Figure 3: *(Left)* Random initialization yields diffuse, inconsistent patterns for rare codes (broad orange peak near 0.75), whereas PLANT restores consistency (sharp orange peak at 0.985). *(Right)* Rare-F1 when training only on common labels (> 1% frequency). PLANT retains strong zero-shot performance (7.3–8.1%); random attention initialization collapses (0.5–1.1%). See Section 3 (RQ6).

Our **main contributions** are as follows: (1) We introduce PLANT (**P**retrained and **L**everaged **A**tte**NT**ion), a plug-and-play strategy for initializing attention. PLANT replaces random initialization with relevance-guided attention weights via a two-stage framework: Stage 1 pre-trains the attention layer as a *Learning-to-Rank* (L2R) module using *mutual information*; Stage 2 leverages these weights to train the full model end-to-end, improving rare-label performance. PLANT is architecture-agnostic and can be seamlessly integrated with LLM backbones — such as `Mistral-7B`, `LLaMA3-8B`, `DeepSeek-V3`, or `Phi-3`— without any modification; (2) In extensive experiments across ICD coding, legal topic classification, and content recommendation, we report consistent gains using PLANT across backbones and datasets, and we analyze through careful ablations which aspects of PLANT are responsible for these.

## 2 PLANT

In Extreme Multilabel Classification (XMTC) tasks, the goal is to assign to an input text multiple relevant labels from a very large label set. Formally, denote the dataset by $\mathcal{D} = \left\{ (\boldsymbol{x}_i, \boldsymbol{y}_i) \mid \boldsymbol{y}_i \in \{0,1\}^{|\mathcal{L}|}, i = 1, \ldots, N \right\}$, where: $\boldsymbol{x}_i$ is an input instance (e.g., a text document), and $\boldsymbol{y}_i$ is a binary vector indicating the presence ($y_{il} = 1$) or absence ($y_{il} = 0$) of each label $l \in \mathcal{L}$, where $\mathcal{L}$ denotes the label set (which may contain tens of thousands of unique labels). The objective is to learn a prediction function $f_\theta : \boldsymbol{x}_i \mapsto \mathbb{R}^{|\mathcal{L}|}$ parameterized by $\theta$ that outputs labels for each input $\boldsymbol{x}_i$. For each label $l \in \mathcal{L}$, the output $f_\theta(\boldsymbol{x}_i)_l \in \mathbb{R}$ is the score assigned for the $l$-th label.

**Model Architecture.** We start with a pretrained transformer-based LLM $\mathcal{M}_{\mathsf{base}}$, selected from widely used models such as `Mistral-7B`, `LLaMA3-8B`, `DeepSeek-V3`, and `Phi-3`, known for strong general (Team et al., 2023; Grattafiori et al., 2024; Jiang et al., 2024; Abdin et al., 2024) and domain-specific performance in ICD coding (Yang et al., 2022a; 2023c; Falis et al., 2024; Madan et al., 2024; Nerella et al., 2024; Asensio Blasco et al., 2025; He et al., 2025; Liu et al., 2025a; Yuan et al., 2025). Due to computational constraints, we use both Low-Rank Adaptation (LoRA) and 4-bit quantization in all experiments (Frantar et al., 2022; Hu et al., 2022; Dettmers et al., 2023; Liu et al., 2024b; Aidouni, 2024). We adapt $\mathcal{M}_{\mathsf{base}}$ into $\mathcal{M}_{\mathsf{adapt}}$; see Appendix A for details.

Given an input $\boldsymbol{x}_i$, we tokenize it into $\boldsymbol{t}_i$ and pass it through the adapted model $\mathcal{M}_{\mathsf{adapt}}$ to obtain hidden states: $\boldsymbol{H}_i = \mathcal{M}_{\mathsf{adapt}}(\boldsymbol{t}_i) \in \mathbb{R}^{n \times h}$, where $n$ is the sequence length and $h$ the hidden size. We extract the final token's representation $\boldsymbol{h}_n \in \mathbb{R}^h$ for label prediction.

Specifically, we define trainable label embeddings $\boldsymbol{E} \in \mathbb{R}^{|\mathcal{L}| \times h}$, one per label. These embeddings serve as query vectors in a multi-head attention module. The module, denoted as MultiHead, defines learnable parameters $\boldsymbol{W}_{\mathsf{attn}}$:

$$\boldsymbol{Q} = \boldsymbol{E}, \ \boldsymbol{K} \,\&\, \boldsymbol{V} = \boldsymbol{H}_i, \quad \boldsymbol{A}, \boldsymbol{S} = \mathsf{MultiHead}^1(\boldsymbol{Q}, \boldsymbol{K}, \boldsymbol{V}; \boldsymbol{W}_{\mathsf{attn}}), \quad \boldsymbol{A} \in \mathbb{R}^{|\mathcal{L}| \times h}, \ \boldsymbol{S} \in \mathbb{R}^{|\mathcal{L}| \times n}, \tag{1}$$

where $\boldsymbol{Q}$, $\boldsymbol{K}$, and $\boldsymbol{V}$ represent the query, key, and value inputs, respectively. During training, the parameters $\boldsymbol{W}_{\mathsf{attn}}$ are optimized to learn label-specific attention weights $\boldsymbol{S}$, which determine how each label query attends to tokens in the sequence, and the output $\boldsymbol{A}$, which represents the attended representations for each label query.

The resulting attention output is boosted by learnable matrices $\mathbf{B}_a, \mathbf{B}_m \in \mathbb{R}^{|\mathcal{L}| \times h}$ as: $\mathbf{A}_{\mathsf{boost}} = \mathbf{B}_a + \mathbf{A} \cdot \mathbf{B}_m$, where $\cdot$ denotes element-wise multiplication. This "boosts" label-specific signals in a learned, differentiable manner. This is motivated by the need to enhance task-specific signals in recent Mixture-of-Experts (MoE) frameworks Cai et al. (2024); Yu et al. (2024); Chen et al. (2023b). An adaptive average pooling layer reduces the dimensionality of $\mathbf{A}_{\mathsf{boost}}$ to: $\boldsymbol{P}_i = \mathsf{Pool}(\mathbf{A}_{\mathsf{boost}}) \in \mathbb{R}^{|\mathcal{L}| \times p}$. [1] A shared linear projection $\boldsymbol{W}_c \in \mathbb{R}^{p \times 1}$ then computes the final logits as $\hat{\boldsymbol{y}}_i = \boldsymbol{P}_i \boldsymbol{W}_c \in \mathbb{R}^{|\mathcal{L}|}$, each entry in $y_i$ is the predicted (raw) relevance score for the corresponding label.

### TWO-STAGE TRAINING

PLANT entails a two stage optimization strategy. The first focuses on pretraining MultiHead (Equation 1) label-wise attention weights; the second entails fine-tuning the model end-to-end.

**STAGE 1: PRETRAINING ATTENTION AS L2R (FIGURE 1)** In Stage 1 we train the multi-head attention module parameters ($\mathcal{M}_{\mathsf{adapt}}$, $\mathbf{E}$ and MultiHead). The MultiHead module outputs label-specific attention scores $\boldsymbol{S} = [s_{lj}] \in \mathbb{R}^{|\mathcal{L}| \times n}$, where $s_{lj}$ is the attention score for token $j$ with respect to label $l$.

We train this following a learning-to-ranking objective focused on the top-$k$ tokens per label selected by **Mutual Information Gain** (MIG) computed from the training set.[2]

where $\mathbf{r}_l = [r_{lj}] \in \mathbb{R}^n$ represents the ground-truth relevance of tokens for label $l$, derived from the MIG between tokens and labels; $\mathcal{T}_l$ is the set of top-$k$ tokens for label $l$, selected based on $r_{lj}$;

---

[1]See Appendix A for the number of attention heads in Equation 1 and output size $p$ used in pooling.

[2]See Appendix B.1 for details on pre-computing MIG for a corpus.

$$\mathcal{L}_{\text{rank}}^{(i)}(\boldsymbol{S}) = -\sum_{l=1}^{|\mathcal{L}|} \sum_{\substack{j,h \in \mathcal{T}_l \\ r_{lj} > r_{lh}}} |\Delta\text{nDCG@k}|_{jh} \times \log_2 \left(1 + e^{-\sigma(s_{lj} - s_{lh})}\right)^{-1} \qquad (2)$$

wt. for pairwise swap impact

MIG relevance

(differentiable) approx ranking prob.

$|\Delta\text{nDCG@k}|_{jh}$ is the change in nDCG@k after swapping $j$ and $h$ in the predicted ranking, where the predicted ranking is determined by sorting tokens in $\mathcal{T}_l$ by their predicted scores $s_{lj}$ in descending order; and $\sigma$ is the sigmoid scaling factor. We test $k = \{500, 1000, 2000\}$ to evaluate sensitivity.

**PLANTed Attention via MIG Ranking** MIG scores, denoted by $r_{lj}$, quantifies how informative token $t_j$ is for predicting label $l$. Higher $r_{lj}$ indicates stronger relevance of the token to the label. Note that we empirically determined these relevance scores by computing MIG between token occurrences and label assignments across the training corpus (see Appendix B.1). The ranking loss in Equation 2 encourages the MultiHead module to assign higher attention scores ($s_{lj}$) to tokens with greater relevance. It considers token pairs $(j, h)$ where token $j$ is more relevant than token $h$ for a given label $l$, i.e., $r_{lj} > r_{lh}$. The term $\left(1 + e^{-\sigma(s_{lj} - s_{lh})}\right)^{-1}$ approximates the probability that token $j$ is ranked above token $h$. Each pair is weighted by $|\Delta\text{nDCG@k}|_{jh}$, which penalizes incorrect rankings in proportion to their impact on the nDCG@k metric. This loss formulation encourages attention scores to align with the MIG.

STAGE 2: LEVERAGING ATTENTION – FULL TRAINING    In Stage 2 we train the entire model (Section 2) end-to-end. We start with the finetuned $\mathcal{M}_{\text{adapt}}$ and the initialized weights $\mathbf{W}_{\text{attn}}$ and $\mathbf{E}$ from Stage 1. We optimize the model under focal loss with label smoothing and hard negative mining to address label imbalance (Ben-Baruch et al., 2020; Xiong et al., 2023). The detailed formulation of the focal loss is provided in Appendix A (Eq. 3).

To address the challenge of imbalanced labels, where negative labels often dominate, hard negative mining is applied to focus the loss on the most informative examples. This selects all positive labels ($y_{il} = 1$) and the top-$m$ negative labels ($y_{il} = 0$) with the highest predicted probabilities $\sigma(\hat{y}_{il})$, where $m = 1000$. The focal loss is then computed over this selected subset $\mathcal{S}_i \subseteq \mathcal{L}$ by restricting the summation in Eq. 3 to $\mathcal{S}_i$. We refer to this as the *HNM-augmented focal loss*. This approach ensures the model prioritizes learning from difficult negative examples, improving performance on challenging cases. (as shown in Ablation Section 4)

# 3 EXPERIMENTS

**Datasets, Baselines & Implementation:** We evaluated PLANT against SOTA models on the `MIMIC-IV/III-full` datasets, which comprise discharge summaries annotated with ICD-9 and ICD-10 codes, respectively. For few-shot learning, we used `MIMIC-III-rare50` and `MIMIC-III-few` subsets to focus on rare codes. To assess generalizability, we also evaluated PLANT on publicly available legal topic classification (`EURLEX-4K`, over long legal documents) and content recommendation (`WIKI10-31K`, tag prediction for Wikipedia-style texts). For complete training-time, memory, inference, and scalability analyses—as well as dataset descriptions, implementation details, baselines, and evaluation metrics—please refer to App. H, C, D, and E.

*Notation.* ▲/▼ mark significant gains/drops ($\alpha=0.05$, Wilcoxon Demšar 2006; see App. F), shown if the test passes and 95% CI excludes 0. Gains/drops followed by CI in plum. **Bold** = best per metric.

**RQ1: How effective is PLANT 's two-stage across LLM backbones?** Table 2 compares the performance of four LLM backbones (`Mistral-7B`, `LLaMA3-8B`, `DeepSeek-V3`, `Phi-3`)[3] trained end-to-end with *HNM-augmented focal loss* versus PLANT 's two-stage training (Section 2) which, in stage 2, adopts the same *HNM-augmented focal loss*, on `MIMIC-III-full` and `MIMIC-IV-full`. We report the *average absolute gains* obtained by computing the mean difference between each LLM and its PLANT-enhanced counterpart for a given metric. Table 2 highlights the gains from integrating PLANT: green rows show PLANT-enhanced results, with consistent improvements across all metrics and average gains summarized in the last row.

Notably, much smaller models integrated with PLANT outperform significantly larger LLMs used alone. For instance, on `MIMIC-III-full`, `LLaMA3-8B` +PLANT (8B) outperforms `DeepSeek-V3` (336B) by **+1.8** in F1 (Macro) and **+3.0** in P@15. Similarly, `Phi-3` +PLANT (3.8B) surpasses `DeepSeek-V3` by **+2.0** in F1 (Micro) and a substantial **+7.1** in AUC (Macro). This trend persists across `MIMIC-IV-full` as well: `LLaMA3-8B` +PLANT achieves gains of **+3.7** in F1 (Macro) and **+3.5** in P@15 over `DeepSeek-V3`. These results highlight the efficiency of PLANT, which permits smaller models to surpass much larger LLMs across key metrics.

**RQ2: Does PLANT[4] outperform SOTA models on ICD-10 code classification?** Table 3 compares PLANT with SOTA models on `MIMIC-IV-full`. Performance comparison across `MIMIC-III-full` and `MIMIC-III-top50` is provided in Table 11 in Appendix G. On `MIMIC-IV-full`, which exhibits a more skewed label distribution (see Table 8), PLANT demonstrates average gains of **+0.2–1.4**, including a **+0.7** (95% CI: **0.5–1.0**) gain in F1 (Macro) and a **+1.4** (95% CI: **1.0–1.9**) improvement in Precision@8 over SOTA baselines. PLANT 's larger performance gains on `MIMIC-III-full` and `MIMIC-IV-full` for the macro-averaged metrics highlight its effectiveness in addressing label imbalance.

**RQ3: How effective is PLANT on rare labels?** Table 4 evaluates PLANT against SOTA models on `MIMIC-III-few` (labels appearing in fewer than 5 samples) and `MIMIC-III-rare50` (50 most rare labels) subsets of `MIMIC-III-full`. PLANT significantly outperforms all baselines. On `MIMIC-III-few`, PLANT achieves substantial aggregate gains of **+30–49** across F1, Precision, and Recall, including a **+36.1** (95% CI: **30.5–41.7**) gain in F1 (Macro) and a **+48.1** (95% CI: **42.6–54.0**) gain in Recall (Macro). For `MIMIC-III-rare50`, PLANT demonstrates even larger improvements, with average gains of **+9–49** across metrics, notably a **+48.6** (95% CI: **41.2–56.4**) gain in F1 (Macro).

**RQ4: Why PLANT Is Superior to Few-Shot SOTA Models?** Figure 2 (Left) shows that codes with frequencies <10 have near-zero macro-F1 scores, highlighting the challenge of predicting *rare codes*—a problem PLANT aims to address. To evaluate this, we used the `MIMIC-III-few` dataset, which contains 685 codes, each appearing in <5 instances. Figure 2 (Right) focuses on these rare codes, effectively zooming in on the leftmost part of Figure 2 (Left). We present violin plots (with embedded box plots) of macro-F1 distributions for rare codes across three models: Co-Relation (Luo et al., 2024) (mean = 0.054), `Mistral-7B` (mean = 0.308), and `Mistral-7B` +PLANT (mean = **0.663**). Notably, **54.8%** of rare codes achieve macro-F1 > 0.7 with PLANT, compared to only 2.0% for the base `Mistral-7B`, and 0.6% for Co-Relation. These results demonstrate that integrating PLANT with a base LLM not only surpasses specialized few-shot approaches but also markedly enhances the LLM's capacity to model rare labels.

**RQ5: How generalizable is PLANT[5] to other imbalanced classification tasks?** Table 5 evaluates PLANT on two diverse tasks: legal topic classification (`EURLEX-4K`) and content recommendation (`WIKI10-31K`), both characterized by extreme label spaces and imbalanced distributions. On `EURLEX-4K`, PLANT achieves aggregate gains of **+0.9–2.5** across P@1, P@3, and P@5, including a a **+2.5** (95% CI: **1.7–3.5**) gain in P@3. For `WIKI10-31K`, PLANT shows a **+2.2** (95% CI: **1.6–2.8**) gain in P@3, though it exhibits a negligible dip in P@5.

**RQ6: How effective is PLANT on zero-shot transfer to unseen rare labels?** Across 4 LLMs, PLANT trained on the *full* dataset achieves the strongest Rare-F1 (14.0–16.8%). Figure 3 (right) reports Rare-F1 when the model is trained exclusively on documents containing only common labels and evaluated on held-out rare labels. The **second and third batches of bars** in the figure correspond to models trained only on the common-label subset: here, PLANT still retains substantial performance (7.3–8.1%), whereas the same models with **random atten-init** collapse to 0.5–1.1% Rare-F1. This gap—up to +15.7 pp for `LLaMA3-8B` —shows that Stage 1 attention initialization enables true zero-shot generalization to unseen rare labels, despite having no rare-label supervision.

---

[3]See Appendix A for LLM details.

[4]In this setting the base LLM $\mathcal{M}_{base}$ for PLANT was `Mistral-7B`.

[5]In this setting the base LLM $\mathcal{M}_{base}$ for PLANT was DistilBERT.

| Model | AUC | | F1 | | P@15 |
|---|---|---|---|---|---|
| | Macro | Micro | Macro | Micro | |
| `Mistral-7B` | 90.2 | 98.7 | 20.0 | 57.0 | 53.8 |
| `Mistral-7B` + PLANT | 97.4▲ *(+7.2)* | 99.5▲ *(+0.8)* | 23.0▲ *(+3.0)* | 59.2▲ *(+2.2)* | 56.9▲ *(+3.1)* |
| `LLaMA3-8B` | 90.5 | 98.8 | 20.5 | 57.5 | 54.0 |
| `LLaMA3-8B` + PLANT | **97.6**▲ *(+7.1)* | **99.6**▲ *(+0.8)* | **23.5**▲ *(+3.0)* | **59.5**▲ *(+2.0)* | **57.0**▲ *(+3.0)* |
| `DeepSeek-V3` | 90.0 | 98.6 | 19.8 | 56.8 | 53.5 |
| `DeepSeek-V3` + PLANT | 97.2▲ *(+7.2)* | 99.4▲ *(+0.8)* | 22.8▲ *(+3.0)* | 59.0▲ *(+2.2)* | 56.5▲ *(+3.0)* |
| `Phi-3` | 89.8 | 98.5 | 19.5 | 56.5 | 53.2 |
| `Phi-3` + PLANT | 97.0▲ *(+7.2)* | 99.3▲ *(+0.8)* | 22.5▲ *(+3.0)* | 58.8▲ *(+2.3)* | 56.3▲ *(+3.1)* |
| **Avg. gain with PLANT** | Δ+7.2 | Δ+0.8 | Δ+3.0 | Δ+2.2 | Δ+3.1 |

Table 2: **PLANT consistently boosts all LLM backbones on `MIMIC-IV-full`.** The full table with both `MIMIC-III-full` and `MIMIC-IV-full` results is provided in Table 10 in Appendix G. See Appendix G Table 12 for propensity scores.

| Model | AUC | | F1 | | Precision | |
|---|---|---|---|---|---|---|
| | Macro | Micro | Macro | Micro | P@8 | P@15 |
| CoRelation (Luo et al., 2024) | 97.2 | 99.6 | 6.3 | 57.8 | 70.0 | 55.3 |
| PLM-CA (Edin et al., 2024) | 91.8 | 99.1 | 22.3 | 58.9 | 70.5 | 55.8 |
| GKI-ICD (Zhang et al., 2025) | 97.1 | 99.3 | 20.6 | 58.5 | 70.7 | 55.8 |
| GPT-4 Zero-Shot (Yuan et al., 2025) | 90.5 | 98.8 | 5.0 | 56.0 | 68.0 | 53.5 |
| PLANT (Ours) | **97.4** | 99.5 | **23.0** | **59.2** | **72.1** | **56.9** |
| | ▲ +0.2 | *-0.1* | ▲ +0.7 | ▲ +0.3 | ▲ +1.4 | ▲ +1.1 |
| | [0.08, 0.31] | [-0.27, 0.06] | [0.48, 0.95] | [0.12, 0.44] | [1.01, 1.88] | [0.72, 1.55] |

Table 3: **PLANT sets a new SOTA on `MIMIC-IV-full`.**

## 4 ABLATION ANALYSIS

**ABLATING COMPONENTS IN PLANT (TABLE 6)** To assess the contribution of individual components in PLANT's two-stage training pipeline (Section 2), we perform ablations on two dataset/LLM combinations: `MIMIC-III-full` with `Mistral-7B` and `MIMIC-IV-full` with `LLaMA3-8B`, where `Mistral-7B` and `LLaMA3-8B` are the base LLMs described in Section 2. **Each ablation configuration is compared against the full PLANT setup** (bottom row of Table 6), intentionally isolating the effect of a component. We evaluate using (1) macro-AUC for per-label classification, (2) macro-F1 for rare-label accuracy, and (3) P@15 for top-$k$ prediction quality. For each ablation, we report the average decrease relative to the full PLANT setup ($\nabla{-}x$).

**(1)+(2) Benefit of Stage 1 attention initialization.** End-to-end training from scratch without Stage 1 degrades performance: using BCE loss yields avg dips (macro-AUC/macro-F1/P@15) of $\nabla{-}6.2$/$\nabla{-}4.8$/$\nabla{-}5.3$, while Focal+HNM ($\gamma = 2, \epsilon = 0.1, m = 1000$) reduces the dips to $\nabla{-}4.8$/$\nabla{-}3.4$/$\nabla{-}3.8$. *Takeaway:* Random attention initialization and label imbalance severely harm rare-label accuracy and top-$k$ retrieval; Focal+HNM helps, but pretrained attention still recovers substantial headroom.

**(3) Removing focal loss and HNM in Stage 2.** Training with vanilla BCE after Stage 1 ($\gamma = 0$, no label smoothing, no HNM), yields changes of (macro-AUC/macro-F1/P@15: $\nabla{-}2.5$/$\nabla{-}2.0$/$\nabla{-}2.5$; *Takeaway:* While not competitive with the full PLANT, it still outperforms both single-stage BCE and single-stage Focal Loss, demonstrating *Stage 1 attention pretraining alone provides meaningful gains* even with simple BCE.

**(4) Effect of label smoothing in Stage 2.** Two-stage training without label smoothing (Stage 2 with $\epsilon{=}0$) results in (macro-AUC/macro-F1/P@15): $\nabla{-}0.3$/$\nabla{-}1.1$/$\nabla{-}1.2$; *Takeaway:* Mild but consistent loss—overconfidence slightly hurts macro-F1.

**(5) Effect of hard negative mining (HNM) in Stage 2.** Two-stage training but without HNM (in Stage 2 computing loss over all labels instead of top-$m$ negatives.) changes performance (macro-

| Model | F1 | | Precision | | Recall | |
|---|---|---|---|---|---|---|
| | Ma. | Mi. | Ma. | Mi. | Ma. | Mi. |
| MSMN + Contrastive (Lu et al., 2023) | 4.3 | 8.5 | 4.5 | **70.9** | 4.2 | 4.5 |
| GP (Yang et al., 2023b) | 30.2 | 35.3 | 27.9 | 38.5 | 32.9 | 32.6 |
| Tr-EHR (Yang et al., 2023c) | 22.0 | 32.5 | 20.5 | 52.0 | 23.5 | 24.0 |
| CoRelation (Luo et al., 2024) | 25.0 | 34.0 | 23.5 | 50.5 | 26.5 | 27.0 |
| PLM-CA (Edin et al., 2024) | 26.5 | 35.0 | 24.5 | 51.5 | 28.0 | 28.5 |
| GKI-ICD (Zhang et al., 2025) | 24.0 | 33.5 | 22.5 | 49.0 | 25.5 | 26.0 |
| PLANT (Ours) | **66.3** | **71.0** | **65.1** | 68.6 | **81.0** | **81.7** |
| | ▲ +36.1 | ▲ +35.7 | ▲ +37.2 | ▼ -2.3 | ▲ +48.1 | ▲ +49.1 |
| | [30.5, 41.7] | [29.8, 41.2] | [31.0, 43.5] | [-3.7, -1.0] | [42.6, 54.0] | [43.3, 54.8] |

Table 4: **Performance on rare labels (`MIMIC-III-few`).** PLANT offers large gains on most metrics. The full table including `MIMIC-III-rare50` appears in Table 13 in Appendix G.

| Model | Legal Topic Classification (`EURLEX-4K`) | | | Content Recommendation (`WIKI10-31K`) | | |
|---|---|---|---|---|---|---|
| | P@1 | P@3 | P@5 | P@1 | P@3 | P@5 |
| XRR (Xiong et al., 2023) | 87.96 | 78.88 | 68.52 | 89.54 | 85.38 | 81.34 |
| X-Transformer w/ RDE (Shi et al., 2024) | 84.60 | 72.61 | 61.35 | 86.15 | 76.99 | 68.75 |
| MatchXML (Ye et al., 2024) | 88.12 | 75.00 | 62.22 | 89.30 | 80.45 | 70.89 |
| DE (Gupta et al.) | 87.60 | 74.39 | 67.80 | 88.21 | 80.29 | 69.91 |
| InceptionXML (Kharbanda et al., 2023) + GANDALF (Kharbanda et al., 2024) | 86.98 | 75.89 | 68.78 | 88.76 | 80.32 | 69.89 |
| CG (Chai et al., 2024) | 87.82 | 76.71 | 68.42 | 87.29 | 79.81 | 68.45 |
| PLANT (Ours) | **90.61** | **81.35** | **70.24** | **90.91** | **87.61** | 81.33 |
| | ▲ +2.20 | ▲ +2.47 | ▲ +0.90 | ▲ +1.37 | ▲ +2.23 | -0.01 |
| | [1.47, 3.25] | [1.73, 3.49] | [0.38, 1.55] | [0.76, 1.93] | [1.60, 2.84] | [-0.27, 0.19] |

Table 5: **PLANT performs strongly across domains**—legal topic classification & tag prediction.

AUC/macro-F1/P@15): ▽−0.6/▽−1.9/▽−2.8; *Takeaway:* Noticeable drops in both precision and rare-label accuracy—HNM focuses learning on informative negatives.

**(6) Importance of MIG vs. naive frequency in Stage 1.** Replacing MIG relevance scores with normalized token frequency per label yields (macro-AUC/macro-F1/P@15): ▽−1.0/▽−2.4/▽−3.3; *Takeaway:* Coarser relevance signals harms rare-label and top-$k$ accuracy.

**(7) Importance of ranking objective in Stage 1.** Compared to full PLANT, replacing Stage 1 pairwise ranking loss (Eq. 2) with MSE, yields *avg dips* (macro-AUC/macro-F1/P@15): ▽−1.5/▽−2.9/▽−3.5; *Takeaway:* The pairwise ranking loss is central to "planting" attention weights that reflect MIG-derived token relevance. Replacing it with MSE removes the ranking signal, leading to weaker alignment between learned attention scores and true relevance, which in turn degrades rare-label accuracy & top-$k$ precision.

**(8) Effect of attention initialization quality of Stage 1.** We vary the number of Stage 1 epochs (1–10) before switching to Stage 2, and measure attention initialization quality at the end of Stage 1 via nDCG@$k$ (computed against MIG relevance as ground truth), alongside final macro-F1 and P@15 after Stage 2 (Figure 4, *left*). Compared to the full PLANT setup (10 epochs), training for only 1 epoch yields *avg dips* (nDCG@$k$/macro-F1/P@15): ▽−0.24/▽−2.4/▽−2.6. As Stage 1 training length increases, nDCG@$k$ steadily improves (e.g., 0.68→0.94 on `MIMIC-IV-full`/`LLaMA3-8B`), and final metrics rise accordingly, saturating at the full PLANT performance. The inset shows when attention weights $\mathbf{W}_{attn}$ is randomly initialized, i.e., no Stage 1 (nDCG@k ≈ 0.05), macro-F1 drops to 13.0%($\Delta = +10.0$ pp gain from full PLANT). Figure 5 reports the same trend for rare-F1 (labels with frequency $< 0.1\%$), which converges to 16.8% under full PLANT. The random-init (inset) yields only 5.5% Rare-F1 ($\Delta = +11.3$ pp), confirming that poor attention initialization is the dominant cause of rare-label poor perfor-

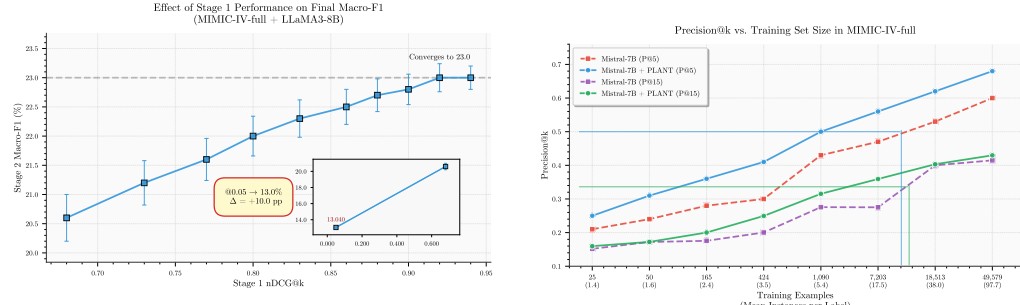

Figure 4: *(Left)* On `MIMIC-IV-full` with `LLaMA3-8B`, better Stage 1 nDCG@$k$ (attn. init. quality) leads to higher Stage 2 (downstream) macro-F1. The same trend for rare-F1 is shown in App. G, Fig. 5. Extended results (`MIMIC-III-full` +`Mistral-7B`, `MIMIC-IV-full` +`LLaMA3-8B`; macro-F1, P@15) appear in App. G, Fig. 6. *(Right)* PLANT consistently boosts `Mistral-7B` on `MIMIC-IV-full` across training sizes: solid lines (`Mistral-7B` +PLANT) beat dashed baselines on P@5/P@15, with largest gains in low-data regimes. Paired `MIMIC-III-full` +`MIMIC-IV-full` results are in App. G, Fig. 7.

| Ablation Config | macro-AUC | macro-F1 | P@15 |
|---|---|---|---|
| [1]Single-Stage BCE | 91.0▼*(-6.4, [-7.2, -5.4])* | 18.0▼*(-5.0, [-5.9, -3.8])* | 52.0▼*(-4.9, [-5.7, -3.9])* |
| [1]Single-Stage Focal Loss | 92.5▼*(-4.9, [-5.8, -4.0])* | 19.5▼*(-3.5, [-4.2, -2.7])* | 53.5▼*(-3.4, [-4.0, -2.7])* |
| [1]PLANT w/ Vanilla BCE | 95.0▼*(-2.4, [-3.0, -1.7])* | 21.0▼*(-2.0, [-2.6, -1.4])* | 54.8▼*(-2.1, [-2.7, -1.5])* |
| [2]PLANT w/o Label Smoothing | 97.0▼*(-0.4, [-0.7, -0.2])* | 21.8▼*(-1.2, [-1.8, -0.7])* | 55.8▼*(-1.1, [-1.7, -0.6])* |
| [2]PLANT w/o Hard Neg Mining | 96.8▼*(-0.6, [-1.0, -0.3])* | 21.0▼*(-2.0, [-2.7, -1.3])* | 54.5▼*(-2.4, [-3.1, -1.7])* |
| [1]PLANT w/ Term Frequency | 96.5▼*(-0.9, [-1.4, -0.5])* | 20.5▼*(-2.5, [-3.2, -1.8])* | 54.0▼*(-2.9, [-3.6, -2.1])* |
| [1]PLANT w/ MSE | 96.0▼*(-1.4, [-2.1, -0.8])* | 20.0▼*(-3.0, [-3.9, -2.1])* | 53.8▼*(-3.1, [-3.9, -2.3])* |
| PLANT (full setup) | **97.4** | **23.0** | **56.9** |

Table 6: Ablation on `MIMIC-IV-full` with base `LLaMA3-8B`. Full results (`MIMIC-III-full` w/ `Mistral-7B`, `MIMIC-IV-full` w/ `LLaMA3-8B`) appear in Table 14, App. G.

mance. *Takeaway:* Higher nDCG@$k$ at the end of Stage 1 correlates (Pearson $r(\mathrm{df}) = .80$, $p < .001$) with better rare-label accuracy and top-$k$ precision.

**Key Takeaways (1–8):** Illustrated in Table 6, Stage 1 attention pretraining is the single most impactful component of PLANT: removing it and training end-to-end from scratch with BCE or HNM-augmented focal loss yields average dips of $\nabla - (4.8 \text{ to } 6.2)$ in macro-AUC, $\nabla - (3.4 \text{ to } 4.8)$ in macro-F1, and $\nabla - (3.8 \text{ to } 5.3)$ in P@15. Analysis of attention initialization quality (Fig. 6) further shows that stronger token-ranking quality at the end of Stage 1 correlates with better downstream macro-F1 and P@15. Within Stage 1, both the MIG relevance signal and the ranking objective are essential for effectively "planting" attention weights: replacing MIG with token frequency causes average dips up to $\nabla - 3.3$ (P@15), and replacing the ranking loss with MSE causes up to $\nabla - 3.5$ (P@15). Moreover, when Stage 2 is trained with vanilla BCE, PLANT still achieves average gains of $\Delta + (1.4 \text{ to } 2.3)$ (macro-AUC/macro-F1/P@15) over end-to-end training with HNM-augmented focal loss, underscoring that *planted attention* alone contributes substantial improvements.

**PLANT UNDER VARYING TRAINING SIZE** Annotated data is scarce and costly, especially for rare labels. So we ask: *can PLANT's pretrained attention improve sample efficiency over standard end-to-end training by reducing the labeled examples needed for competitive performance?* To evaluate this, we compare PLANT 's two-stage training (Section 2) with single-stage end-to-end training using the same architecture (Section 2) and base LLM `Mistral-7B`, on `MIMIC-III-full` and `MIMIC-IV-full` under varying training sizes (Figure 4, right). Both methods are trained on different fractions of balanced training splits, with fixed test sets, up to 5 epochs, and evaluated on P@5 and P@15. The sole difference is in the attention mechanism **A** from the MultiHead module (Equation 1): PLANT uses Stage 1 pretrained attention from the L2R model (Equation 2), emphasizing MIG-ranked tokens, while the baseline learns attention from scratch. **Takeaway:** As shown in Figure 4 (right), PLANT consistently matches or exceeds end-to-end `Mistral-7B` across all training sizes, often with an order of magnitude fewer labels. On `MIMIC-IV-full`, PLANT

| Variant (training data) | Macro-F1 (%) | Rare-F1 (%) | Corr(F1, log-freq) | Sim($\mathbf{S}_l$, MIG)-Rare |
|---|---|---|---|---|
| Random Attn Init (full data) | $13.0 \pm 1.1$ | $5.5 \pm 0.9$ | $0.68 \pm 0.04$ | $0.12 \pm 0.06$ |
| 1 Epoch Stage 1 PLANT (full data) | $20.5 \pm 0.6$ | $10.3 \pm 0.9$ | $0.55 \pm 0.05$ | $0.35 \pm 0.07$ |
| PLANT (full data) | $23.0 \pm 0.3$ | $16.8 \pm 0.5$ | $0.30 \pm 0.03$ | $0.78 \pm 0.04$ |
| Random Attn Init (common only) | $12.5 \pm 1.0$ | $0.5 \pm 0.4$ | $0.78 \pm 0.03$ | $0.08 \pm 0.05$ |
| PLANT (common only) | $18.0 \pm 0.7$ | $8.0 \pm 0.6$ | $0.45 \pm 0.04$ | $0.52 \pm 0.07$ |

Table 7: Impact of PLANT on rare-label performance in `MIMIC-IV-full` using `LLaMA3-8B`.

achieves P@5=0.50 and P@15=0.37 with only **1090** and **2743** instances—matching baselines trained on 10,337 and 12,902. On `MIMIC-III-full`, it reaches P@5=0.47 and P@15=0.30 using just **136** and **235** instances—vs. 1342 and 1578 for the baseline.

**RANDOM ATTENTION INITIALIZATION CAUSES RARE-LABEL FAILURES**   In Table 7 [6], random initialization yields the **highest frequency–F1 correlation** (0.68–0.78), indicating strong bias toward frequent labels, and the **lowest alignment** between learned attention scores $\mathbf{S}_l$ and ground-truth MIG relevance profiles. Alignment is measured via **cosine similarity** between $\mathbf{S}_l$ and the MIG vector for each rare label: PLANT reaches 0.78 (attention concentrated on truly informative tokens), whereas random initialization collapses to 0.08–0.12 (diffuse & uninformative). Comparisons show PLANT's attention initialization mitigates both frequency bias and attention misalignment.

## 5   RELATED WORK

Attention has long been used to capture label–text interactions. You et al. (2019) used bi-LSTMs and a label-tree–guided attention mechanism to produce label-specific representations. Transformer-based models introduced multi-resolution self-attention for large label spaces (Zhang et al., 2021; Kharbanda et al., 2022), while multi-head attention across text granularities improved weak supervision (Kargupta et al., 2023). also leveraging contrastive or knowledge-enhanced attention (Lu et al., 2023; Li et al., 2023). Dynamic pipelines filtered candidate labels using structured signals like diagnoses, procedures, and medications, relying on attention to prioritize relevant labels (Wang et al., 2024b;c). Other studies show that label-guided, dictionary, or bi-attention mechanisms improve alignment between labels and text (Wang et al., 2023b; Wu et al., 2024; Wang et al., 2024a). Meta-learning and label tree structures further advance attention-driven few-shot generalization (Teng et al., 2024; Wang et al., 2024d). Recent work includes attention-based co-ranking (Yan et al., 2025), contrastive dual-attention for rare labels (Huang et al., 2025), and knowledge-integrated attention for medical coding (Zhang et al., 2025). PLANT is the first work to *pretrain attention*.

LLMs are increasingly used for XMTC (Asensio Blasco et al., 2025; Yuan et al., 2025; Nerella et al., 2024). Yet large LLMs in zero-shot mode can underperform smaller fine-tuned models (Boyle et al., 2023; Zhang et al., 2025). Heavy finetuning, in turn, raises concerns about compute cost and overfitting (Huang et al., 2022; Michalopoulos et al., 2022; Ng et al., 2023; Kang et al., 2023). Sakai & Lam (2025) further show that such finetuning often fails to improve rare-label performance in high-dimensional, skewed label spaces. As PLANT is *architecture-agnostic and effective under skew*, it integrates seamlessly with LLMs to boost rare-label performance without heavy finetuning.

## 6   CONCLUSION

This work proposed PLANT—a plug-and-play strategy for initializing attention. By pretraining attention as a L2R module with mutual information and then leveraging it in full end-to-end training, PLANT turns attention into a pretrainable component. PLANT is architecture-agnostic, integrates seamlessly with diverse LLM backbones, and boosts performance across tasks. Strikingly, smaller LLMs enhanced with PLANT outperform much larger models used alone, and PLANT is substantially better at predicting rare labels. It also improves sample efficiency, matching the performance of baselines trained on $10\times$ more data. In sum, PLANT shifts attention from something merely learned during training to something we can *plant* and leverage. Looking ahead, its pretraining principle could extend naturally to multimodal tasks, where cross-signal attention is critical.

---

[6]Full ablation setup and experimental details are provided in Appendix A:CAUSAL ABLATION DETAILS.

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

## A  TRAINING DETAILS

**LLM Backbone**  We use the following pretrained instruction-tuned LLMs as base models $\mathcal{M}_{\text{base}}$ in our experiments, all publicly available on the Hugging Face Model Hub and compatible with the Transformers library: (1) **Mistral-7B-Instruct-v0.3** (7B, Mistral AI): https://huggingface.co/mixtral-7b-instruct-v0.3, (2) **Llama-3.1-8B** (8B, Meta AI): https://huggingface.co/meta-llama/Llama-3.1-8B, (3) **DeepSeek-R-336B** (336B, DeepSeek): https://huggingface.co/deepseek/DeepSeek-R-336B, and (4) **Phi-3-mini-3.8B** (3.8B, Microsoft): https://huggingface.co/microsoft/Phi-3-mini-3.8B. These serve as the LLM backbones for fine-tuning.

For the extreme multi-label text classification (XMTC) results reported in Table 5, PLANT was additionally adapted to a compact **DistilBERT** encoder backbone (66M parameters) to ensure a fair comparison with the listed baselines (e.g., XRR (Xiong et al., 2023), MatchXML (Ye et al., 2024), and InceptionXML (Kharbanda et al., 2023)), which similarly employ encoder models of comparable scale. The DistilBERT model is initialized from `distilbert-base-uncased` and is publicly accessible at https://huggingface.co/distilbert/distilbert-base-uncased.

**Quantization & LoRA Adaptation**  Starting with a pretrained model $\mathcal{M}_{\text{base}}$, such as `Mistral-7B`, `LLaMA3-8B`, or `Phi-3`, we apply Parameter-Efficient Fine-Tuning (PEFT) using Low-Rank Adaptation (LoRA) Hu et al. (2022); Dettmers et al. (2023); Liu et al. (2024b).

To enable memory-efficient fine-tuning on resource-constrained hardware, we first quantize $\mathcal{M}_{\text{base}}$ to 4-bit precision using the `NormalFloat4` format with double quantization, yielding $\mathcal{M}_{\text{quant}}$:

$$Q(\boldsymbol{W}) = \text{round}\left(\frac{\boldsymbol{W}}{s}\right) \cdot s,$$

where $\boldsymbol{W}$ is a model weight matrix and $s$ is a learned scale. Inference is performed using `bfloat16` precision ($\mathbb{F}_{16}b$) (Refer to Frantar et al. (2022) for details).

We then apply LoRA to a subset of the attention projection layers (query, key, value, and output), introducing trainable low-rank matrices:

$$\Delta \boldsymbol{W} = \boldsymbol{AB}, \quad \text{with } \boldsymbol{A} \in \mathbb{R}^{d \times r}, \boldsymbol{B} \in \mathbb{R}^{r \times d},$$

using rank $r = 16$, scaling factor $\alpha = 32$, and dropout $p = 0.05$. The adapted model becomes:

$$\mathcal{M}_{\text{adapt}} = \mathcal{M}_{\text{quant}} + \alpha \cdot \Delta \boldsymbol{W}.$$

**Optimization and Training Regimen**  To address potential overwriting of Stage 1 attention signals during Stage 2 fine-tuning, we employ a gradual unfreezing strategy combined with discriminative learning rates, ensuring stable transfer of the MIG-seeded priors while allowing task-specific refinement. All experiments are conducted on $8 \times$A100–80GB GPUs using DeepSpeed ZeRO-3 offloading for memory efficiency, with a global batch size of 256 (gradient accumulation steps=4) and mixed-precision (FP16) training via Hugging Face Accelerate.

Stage 1 pretraining optimizes the multi-head attention module (MultiHead) and label embeddings $\mathbf{E}$ via the ranking loss (Eq. 2) for 10 epochs, using AdamW with a cosine learning-rate schedule (peak $\eta = 5 \times 10^{-4}$, 10% warmup) and weight decay $\lambda = 0.01$.

In Stage 2, we initialize from Stage 1 checkpoints and apply discriminative fine-tuning to preserve attention integrity: the attention module (MultiHead, $\mathbf{E}$) starts frozen for the first 5 epochs (allowing downstream layers to adapt), followed by gradual unfreezing of the full model in three phases—attention last (epochs 6–10, $\eta = 1 \times 10^{-5}$), intermediate layers (epochs 11–15, $\eta = 5 \times 10^{-6}$), and all parameters (epochs 16–20, $\eta = 2 \times 10^{-6}$)—each with cosine decay and 5% warmup. This layered schedule, inspired by progressive distillation in large-scale vision–language models (Hou et al., 2018), is paired with the AdamW optimizer Loshchilov & Hutter (2017) (weight decay 0.01) and gradient clipping (max-norm 1.0) for stability. We use a per-device batch size of 8 with 4-step gradient accumulation (effective batch size 32), PyTorch's `autocast` for FP16, and gradient checkpointing to manage memory. Each stage runs up to 10 epochs with early stopping (patience=2): validation nDCG@k for Stage 1, macro-F1 for Stage 2. To ensure reproducibility, we fix random seeds across `random`, `numpy`, `torch`, and `torch.cuda`; experiments use distributed data-parallelism (DDP) where applicable, with metrics logged via Weights & Biases.

**Token Selection Sensitivity** To test sensitivity to token selection in the ranking loss (Equation 2), we vary the top-$k$ token threshold with $k \in \{500, 1000, 2000\}$.

### HYPERPARAMETERS IN ARCHITECTURE (SECTION 2)

The multi-head attention module MultiHead (Equation 1) uses $k = 8$ attention heads. The adaptive average pooling layer (Equation 2) produces an output size of $p = 128$.

### FOCAL LOSS WITH LABEL SMOOTHING

For completeness, we present the explicit formulation of the focal loss used in Stage 2 training (Section 2). The focal loss for an input $\mathbf{x}_i$ is defined as:

$$\mathcal{L}_{\text{focal}}^{(i)}(\tilde{y}, \hat{y}, \theta) = -\frac{1}{|\mathcal{L}|} \sum_{l=1}^{|\mathcal{L}|} \left[ \tilde{y}_{il} \left(1 - \sigma(\hat{y}_{il})\right)^{\gamma} \log\left(\sigma(\hat{y}_{il})\right) + (1 - \tilde{y}_{il}) \left(\sigma(\hat{y}_{il})\right)^{\gamma} \log\left(1 - \sigma(\hat{y}_{il})\right) \right], \tag{3}$$

where $\theta$ denotes all trainable model parameters, $\sigma(\cdot)$ is the sigmoid function, $\hat{y}_{il} \in \mathbb{R}$ is the predicted logit for label $l$, $\gamma = 2$ is the focusing parameter that emphasizes harder examples, and $\tilde{y}_{il}$ is the smoothed label: $\tilde{y}_{il} = (1 - \epsilon)^{y_{il}} (\epsilon)^{1 - y_{il}}$ with $\epsilon = 0.1$ to prevent overconfidence in predictions.

### CAUSAL ABLATION DETAILS

To establish that random initialization of the label-specific attention module is the primary cause of poor rare-label performance, we run controlled ablations on MIMIC-IV ICD-10 using LLaMA3-8B. Labels are stratified by training-set frequency: **rare** ($< 0.1\%$) and **common** ($> 1\%$). We evaluate two matched setups: (1) **Random Attn Init (common only)**—Stage 1 skipped; attention weights $\mathbf{W}_{\text{attn}}$ and label embeddings $\mathbf{E}$ remain Xavier-initialized; (2) **PLANT (common only)**—full Stage 1 applied only to the common-label subset. Both models are trained solely on common labels and evaluated on held-out rare labels.

All experiments use 5-fold cross-validation on MIMIC-IV-full (80/10/10 split). Backbone: LLaMA3-8B with QLoRA (rank 16, $\alpha = 32$, 4-bit). Stage 2: 20 epochs, AdamW (lr = 1e$^{-5}$), focal loss ($\gamma = 2$), label smoothing ($\epsilon = 0.1$), and hard-negative mining ($m = 1000$). MIG top-$k = 1000$ is computed on the corresponding training subset. Common-only training removes all rare-label instances. Metrics include micro-F1 per frequency bin and Pearson correlation between per-label F1 and log-frequency. Cosine similarity is averaged over 50 randomly sampled rare labels using full-corpus MIG as reference. All results use seed 42; paired t-tests show $p < 0.001$ for Rare-F1 differences, with Cohen's $d > 1.4$.

## B PRECOMPUTATIONS

### B.1 MUTUAL INFORMATION GAIN IN XMTC

In extreme multilabel classification (e.g., ICD coding on MIMIC-IV), MIG quantifies the informativeness of a token $t_j$ for predicting label $l$ presence, grounded in information theory as the KL

divergence between the joint and product-of-marginals distributions. Formally:

$$r_{l,j} = \sum_{(x,y) \in \{0,1\}^2} P(x,y) \log \left( \frac{P(x,y)}{P(x)\,P(y)} \right),$$

where $x = 1[l \text{ present}]$ (marginal $P(x)$ = label frequency), $y = 1[t_j \text{ present}]$ (marginal $P(y)$ = token frequency), and $P(x,y)$ is the empirical joint from corpus co-occurrences. This measures bits of mutual information: how much $y$ reduces entropy in $x$, penalizing spurious correlations (e.g., high $P(y)$ but low $P(x|y) > P(x)$).

Probabilities are estimated via maximum-likelihood on the full training corpus (no subsampling), with Laplace smoothing ($\alpha = 1$) for zero-count cells to avoid undefined logs. Scores $r_{l,j}$ are L2-normalized per label to $[0, 1]$ (dividing by $\max_j r_{l,j}$) for stability, then thresholded at top-$k$ (tested $k \in \{500, 1000, 2000\}$) to select tokens for Stage 1 ranking. Unlike raw co-occurrence (e.g., $P(y|x)$), MIG corrects for frequency bias: high-frequency tokens inflate joints but are down-weighted if independent of $l$.

**Example.** Consider a toy corpus ($N = 100$ docs): rare label A ($P(x) = 0.05$, 5 docs), common B ($P(x) = 0.50$, 50 docs); token "fever" ($P(y) = 0.40$, co-occurs with B in 25 docs); "rare_disease" ($P(y) = 0.06$, co-occurs with A in 5 docs). Raw co-occurrence ranks "fever" higher for B (support=25 vs. 5), but MIG elevates "rare_disease" for A due to stronger conditional dependence.

For "fever" w.r.t. B, the contingency table yields joints: $P(x = 1, y = 1) = 0.25$, $P(x = 1, y = 0) = 0.25$, $P(x = 0, y = 1) = 0.15$, $P(x = 0, y = 0) = 0.35$. MI computation (log base 2):

$$(1,1): 0.25 \cdot \log_2(0.25/(0.50 \cdot 0.40)) \approx 0.25 \cdot 0.322 = 0.0805,$$
$$(1,0): 0.25 \cdot \log_2(0.25/(0.50 \cdot 0.60)) \approx 0.25 \cdot (-0.263) = -0.066,$$
$$(0,1): 0.15 \cdot \log_2(0.15/(0.50 \cdot 0.40)) \approx 0.15 \cdot (-0.415) = -0.062,$$
$$(0,0): 0.35 \cdot \log_2(0.35/(0.50 \cdot 0.60)) \approx 0.35 \cdot 0.223 = 0.078,$$
$$\text{Total MI} \approx 0.030 \text{ bits (weak dependence)}.$$

For "rare_disease" w.r.t. A: $P(x = 1, y = 1) = 0.05$, $P(x = 1, y = 0) = 0.00$, $P(x = 0, y = 1) = 0.01$, $P(x = 0, y = 0) = 0.94$. MI:

$$(1,1): 0.05 \cdot \log_2(0.05/(0.05 \cdot 0.06)) \approx 0.05 \cdot 4.06 = 0.203,$$
$$(0,1): 0.01 \cdot \log_2(0.01/(0.95 \cdot 0.06)) \approx 0.01 \cdot (-2.51) = -0.025,$$
$$(0,0): 0.94 \cdot \log_2(0.94/(0.95 \cdot 0.94)) \approx 0.94 \cdot 0.0076 = 0.007,$$
$$(1,0): 0 \cdot \cdots = 0,$$
$$\text{Total MI} \approx 0.185 \text{ bits (strong dependence)}.$$

MIG ranks "rare_disease" higher for A ($0.185 > 0.030$), capturing precision ($P(A \mid \text{rare\_disease}) = 83\%$ vs. marginal 5%) without volume bias.

## C  IMPLEMENTATION DETAILS

### DATASETS

We compare PLANT to SOTA ICD coding models using the MIMIC-III (Johnson et al., 2016) and MIMIC-IV (Johnson et al., 2023) datasets, which include rich textual and structured records from ICU settings, primarily discharge summaries annotated with ICD-9 (MIMIC-III) and ICD-10 (MIMIC-IV) codes. MIMIC-III contains 52,722 discharge summaries with 8,929 unique ICD-9 codes, and MIMIC-IV includes 122,279 summaries with 7,942 ICD-10 codes. We follow established methodologies for patient ID-based splits and frequent code subsets. For few-shot learning, we evaluate PLANT on the MIMIC-III-rare50 dataset (Yang et al., 2022b), which features 50 rare ICD codes, and the MIMIC-III-few dataset (Yang et al., 2023b), a subset with 685 unique ICD-9 codes occurring between 1 and 5 times in the training set. We denote these datasets as `MIMIC-III-full`, `MIMIC-III-top50`, `MIMIC-III-rare50`, `MIMIC-III-few`, and `MIMIC-IV-full` (refer to Table 8 for statistics). Following prior research (Mullenbach et al.,

2018; Xie et al., 2019; Li & Yu, 2020), we tokenize and lowercase all text while eliminating non-alphabetic tokens containing numbers or punctuation.

To assess generalizability beyond the clinical domain, we also experiment with two large-scale extreme multilabel datasets. The `EURLEX-4K` dataset, comprising 15,449 training and 3,865 test European Union legal documents annotated with 3,956 EUROVOC labels, supports automated legal topic classification, compliance analysis, and cross-lingual information retrieval (`http://manikvarma.org/downloads/XC/XMLRepository.html`). The `WIKI10-31K` dataset, with 14,146 training and 6,616 test Wikipedia articles associated with 30,938 categories, facilitates automatic tagging, web-scale document organization, and content recommendation (`http://manikvarma.org/downloads/XC/XMLRepository.html`). Both datasets are used to study large-scale label spaces and imbalanced label distributions(refer to Table 9 for statistics).

| | MIMIC-III-full | MIMIC-IV-full |
|---|---|---|
| Number of documents | 52,723 | 122,279 |
| Number of patients | 41,126 | 65,659 |
| Number of unique codes | 8,929 | 7,942 |
| Codes per instance: Median (IQR) | 14(10–20) | 14(9–20) |
| Words per document: Median (IQR) | $1,375(965-1,900)$ | $1,492(1,147-1,931)$ |
| Documents: Train/val/test [%] | 90.5/3.1/6.4 | 72.9/10.9/16.2 |

Table 8: Descriptive statistics for `MIMIC-III-full` and `MIMIC-IV-full` discharge summary training sets.

| | EURLEX-4K | WIKI10-31K |
|---|---|---|
| Number of train documents | $15,449$ | $14,146$ |
| Number of test documents | $3,865$ | $6,616$ |
| Number of unique labels | $3,956$ | $30,938$ |
| Average number of labels per instance | 5.30 | 18.64 |
| Average number of instances per label | 20.79 | 8.52 |

Table 9: Descriptive statistics for publicly available XMTC datasets `EURLEX-4K` and `WIKI10-31K`.

IMPLEMENTATION AND HYPERPARAMETERS

We ensure robustness across diverse XMTC datasets by fine-tuning hyperparameters on the `MIMIC-III-full` and `MIMIC-IV-full` validation sets. Experiments are conducted on an NVIDIA QUADRO RTX 8000 GPU with 48 GB VRAM. We utilize the AWD-LSTM LM with an embedding size of 400, 3 LSTM layers with 1152 hidden activations, and the Adam Optimizer with $\beta_1 = 0.9$, $\beta_2 = 0.99$, and weight decay of 0.01. During fine-tuning, we apply dropout rates and weight dropout, with a batch size of 384, BPTT of 80, 20 epochs, and a learning rate of $1e-5$. Classifier training also includes dropout rates and weight dropout, with a batch size of 16, BPTT of 72, and discriminative fine-tuning with gradual unfreezing over 115 epochs (on `MIMIC-III-full`), alongside scheduled weight decay and learning rate ranges.

# D  BASELINES FOR COMPARISONS

**ICD Baselines:** We compare PLANT against a diverse set of ICD coding baselines spanning classical, recent, and few-shot paradigms.

*Early deep learning models*: CAML (Mullenbach et al., 2018), MSATT-KG (Xie et al., 2019), MUltiResCNN (Li & Yu, 2020), and HyperCore (Cao et al., 2020).

*Attention- and hierarchy-based models:* LAAT and JointLAAT (Vu et al., 2021), ISD (Zhou et al., 2021), Effective-CAN (liu et al., 2021), Hierarchical (Dai et al., 2022), and MSMN (Yuan et al., 2022).

*Recent pretraining and architecture innovations:* DiscNet (Zhang et al., 2022), KEPTLongformer (Yang et al., 2022b), PLM-ICD (Huang et al., 2022), AHDD (Zhang & Wang, 2024), CoRelation (Luo et al., 2024), Contrastive (Lu et al., 2023), MIMIC-IV-Benchmark (Nguyen et al., 2023), Tr-EHR (Yang et al., 2023c), and PLM-CA (Edin et al., 2024).

*Few-shot ICD coding methods:* AGMHT (Song et al., 2021), RareCodes (Chen et al., 2023a), GP (Yang et al., 2023b), and KEPT (Yang et al., 2022b).

*Knowledge-injected models:* KEMTL (Li et al., 2023), MRR (Wang et al., 2024b), AKIL (Wang et al., 2024c), and GKI-ICD (Zhang et al., 2025).

**XMTC Baselines**: We also compare PLANT against XMTC models like: PECOS (Yu et al., 2022), ICXML (Zhu & Zamani, 2023), XRR (Xiong et al., 2023), RDE (Shi et al., 2024), MatchXML (Ye et al., 2024), DE (Gupta et al.), InceptionXML (Kharbanda et al., 2023), GANDALF (Kharbanda et al., 2024), CG (Chai et al., 2024).

## E    EVALUATION METRICS

We focus on micro-F1, macro-F1, micro-P, macro-P, micro-R, macro-R, micro-AUC, macro-AUC, P@k, and R@k to compare with prior ICD studies. Micro-averaging treats each (text, code) pair individually, aggregating true positives, false positives, and false negatives across all instances. Macro-averaging computes metrics per label, giving more weight to infrequent labels. micro-P is the ratio of aggregated true positives to the sum of true positives and false positives, while macro-P averages precision across all labels. micro-R is the ratio of aggregated true positives to the sum of true positives and false negatives, while macro-R averages recall across all labels. micro-AUC computes the area under the ROC curve for all instances aggregated together, while macro-AUC averages the AUC scores across all labels. P@k and R@k measure the proportion of the top $k$ predicted labels that match the ground truth, focusing on precision and recall, respectively.

$$\text{micro-P} = \frac{\sum_i \text{TP}_i}{\sum_i (\text{TP}_i + \text{FP}_i)}$$

$$\text{micro-R} = \frac{\sum_i \text{TP}_i}{\sum_i (\text{TP}_i + \text{FN}_i)}$$

$$\text{micro-F1} = \frac{2 \cdot \sum_i \text{TP}_i}{\sum_i (\text{TP}_i + \text{FP}_i) + \sum_i (\text{TP}_i + \text{FN}_i)}$$

$$\text{micro-AUC} = \int_0^1 \text{TPR}_{\text{micro}}(\text{FPR}_{\text{micro}}) \, d\text{FPR}_{\text{micro}}$$

$$\text{macro-P} = \frac{1}{L} \sum_{i=1}^{L} \frac{\text{TP}_i}{\text{TP}_i + \text{FP}_i}$$

$$\text{macro-R} = \frac{1}{L} \sum_{i=1}^{L} \frac{\text{TP}_i}{\text{TP}_i + \text{FN}_i}$$

$$\text{macro-F1} = \frac{1}{L} \sum_{i=1}^{L} \frac{2 \cdot \text{TP}_i}{\text{TP}_i + \text{FP}_i + \text{TP}_i + \text{FN}_i}$$

$$\text{macro-AUC} = \frac{1}{L} \sum_{i=1}^{L} \int_0^1 \text{TPR}_i(\text{FPR}_i) \, d\text{FPR}_i$$

$$\text{P@}k = \frac{1}{k} \sum_{i=1}^{k} \mathbb{1}\left[\text{pred}_i \in Y\right]$$

$$\text{R@}k = \frac{1}{\min(k, |Y|)} \sum_{i=1}^{k} \mathbb{1}\left[\text{pred}_i \in Y\right]$$

where $\text{TP}_i$, $\text{FP}_i$, and $\text{FN}_i$ are the true positives, false positives, and false negatives for label $i$, respectively, $L$ is the total number of labels, $\text{TPR}_{\text{micro}}$ and $\text{FPR}_{\text{micro}}$ are the true positive rate and false positive rate for the aggregated micro-averaged data, $\text{TPR}_i$ and $\text{FPR}_i$ are the true positive rate and false positive rate for label $i$, $Y$ is the ground truth label set for an instance, and $\text{pred}_i$ is the $i$-th top predicted label.

## F  STATISTICAL SIGNIFICANCE

**Statistical Significance via Wilcoxon Signed-Rank Test.** We assess statistical significance using the non-parametric Wilcoxon Signed-Rank Test (Demšar, 2006) for comparing paired model outputs. For metrics computed at the instance level (e.g., P@15), we apply the test directly to the paired per-instance scores between the base model and its PLANT-enhanced counterpart. For aggregate metrics such as F1 and AUC, which are reported as single values over the full test set, we first collect $N$ paired scores—either from repeated evaluations (e.g., $N = 10$ in 10-fold cross-validation) or from $N$ bootstrap resamples. Let $\{a_1, a_2, \ldots, a_N\}$ and $\{b_1, b_2, \ldots, b_N\}$ denote the scores of the base model and the PLANT-enhanced model, respectively. We compute the difference $d_i = b_i - a_i$ for each pair and rank the absolute values $|d_i|$ (excluding zeros), averaging ranks in the case of ties. Each rank is assigned the sign of $d_i$, and we compute the rank sums $W^+$ and $W^-$ over positive and negative differences. The test statistic is $W = \min(W^+, W^-)$.

For small $N$, statistical significance is determined using exact Wilcoxon critical values; for larger $N$, we apply the normal approximation with

$$\mu = \frac{N(N+1)}{4}, \quad \sigma = \sqrt{\frac{N(N+1)(2N+1)}{24}},$$

$$z = \frac{W - \mu}{\sigma}.$$

We reject the null hypothesis of no difference if the resulting $p$-value is less than a threshold $\alpha$ (typically 0.05). In our tables, statistically significant improvements are marked using ▲. This test is readily implemented in standard libraries such as `scipy.stats.wilcoxon` in Python or `wilcox.test(paired=TRUE)` in R.

**Reporting Gains with Confidence Intervals.** We also report absolute gains along with 95% confidence intervals (CI) using paired bootstrap resampling. For each evaluation metric, we draw $B = 1000$ bootstrap samples from the test set and compute the difference $\Delta_b = \text{Metric}_b^{\text{PLANT}} - \text{Metric}_b^{\text{Base}}$ for each sample $b$. The reported gain is the mean $\hat{\mu}$ of $\{\Delta_b\}$, and the CI is computed using the percentile bootstrap method by taking the 2.5th and 97.5th percentiles of the empirical distribution of $\{\Delta_b\}$.

We mark results as statistically significant only if the Wilcoxon signed-rank test ($\alpha$=0.05) is passed *and* the 95% CI excludes 0. In such cases, we annotate the score with a colored arrow: ▲ for statistically significant gains and ▼ for significant drops. If the CI includes 0, no arrow is shown. For example, $14.7^▲$ (+1.2, [0.6, 1.8]) indicates a statistically significant gain over the base model, while $70.1^▼$(-1.4, [-2.1, -0.7]) denotes a significant drop. In contrast, 73.8 (+0.3, [0.0, 0.6]) is not statistically significant and is shown without an arrow.

# G ADDITIONAL RESULTS

| Model | MIMIC-III-full | | | | | MIMIC-IV-full | | | | |
|---|---|---|---|---|---|---|---|---|---|---|
| | AUC | | F1 | | P@15 | AUC | | F1 | | P@15 |
| | Macro | Micro | Macro | Micro | | Macro | Micro | Macro | Micro | |
| Mistral-7B | 90.8 | 98.9 | 13.5 | 62.0 | 63.5 | 90.2 | 98.7 | 20.0 | 57.0 | 53.8 |
| Mistral-7B + PLANT | 98.1▲ (+7.3) | 99.9▲ (+1.0) | 14.7▲ (+1.2) | 64.1▲ (+2.1) | 65.8▲ (+2.3) | 97.4▲ (+7.2) | 99.5▲ (+0.8) | 23.0▲ (+3.0) | 59.2▲ (+2.2) | 56.9▲ (+3.1) |
| LLaMA3-8B | 91.0 | 99.0 | 13.8 | 62.5 | 64.0 | 90.5 | 98.8 | 20.5 | 57.5 | 54.0 |
| LLaMA3-8B + PLANT | 98.3▲ (+7.3) | 99.8▲ (+0.8) | 15.0▲ (+1.2) | 64.5▲ (+2.0) | 66.2▲ (+2.2) | 97.6▲ (+7.1) | 99.6▲ (+0.8) | 23.5▲ (+3.0) | 59.5▲ (+2.0) | 57.0▲ (+3.0) |
| DeepSeek-V3 | 90.6 | 98.8 | 13.2 | 61.8 | 63.2 | 90.0 | 98.6 | 19.8 | 56.8 | 53.5 |
| DeepSeek-V3 + PLANT | 97.9▲ (+7.3) | 99.7▲ (+0.9) | 14.5▲ (+1.3) | 64.0▲ (+2.2) | 65.5▲ (+2.3) | 97.2▲ (+7.2) | 99.4▲ (+0.8) | 22.8▲ (+3.0) | 59.0▲ (+2.2) | 56.5▲ (+3.0) |
| Phi-3 | 90.4 | 98.7 | 13.0 | 61.5 | 63.0 | 89.8 | 98.5 | 19.5 | 56.5 | 53.2 |
| Phi-3 + PLANT | 97.7▲ (+7.3) | 99.6▲ (+0.9) | 14.3▲ (+1.3) | 63.8▲ (+2.3) | 65.3▲ (+2.3) | 97.0▲ (+7.2) | 99.3▲ (+0.8) | 22.5▲ (+3.0) | 58.8▲ (+2.3) | 56.3▲ (+3.1) |
| **Avg. gain with PLANT** | Δ+7.3 | Δ+0.9 | Δ+1.3 | Δ+2.2 | Δ+2.3 | Δ+7.2 | Δ+0.8 | Δ+3.0 | Δ+2.2 | Δ+3.1 |

Table 10: **Performance of LLMs with and without PLANT.** Each model is evaluated standalone and with PLANT on `MIMIC-III-full` and `MIMIC-IV-full`. Green rows denote results after integrating PLANT. Bold values indicate the best score for each metric. A compact version with only `MIMIC-IV-full` results is provided in Table 2 in the main paper.

| Model | MIMIC-III-full | | | | | | MIMIC-III-top50 | | | | |
|---|---|---|---|---|---|---|---|---|---|---|---|
| | AUC | | F1 | | Precision | | AUC | | F1 | | P@5 |
| | Macro | Micro | Macro | Micro | P@8 | P@15 | Macro | Micro | Macro | Micro | |
| Effective-CAN liu et al. (2021) | 92.1 | 98.9 | 10.6 | 58.9 | 75.8 | 60.6 | 92.0 | 94.5 | 66.8 | 71.7 | 66.4 |
| MSMN Yuan et al. (2022) | 95.0 | 99.2 | 10.3 | 58.4 | 75.2 | 59.9 | 92.8 | 94.7 | 68.3 | 72.5 | 68.0 |
| PLM-ICD (Huang et al., 2022) | 92.6 | 98.9 | 10.4 | 59.8 | 77.1 | 61.3 | 91.0 | 93.4 | 66.3 | 71.9 | 66.0 |
| Contrastive + JointLAAT (Lu et al., 2023) | 94.1 | 98.8 | 11.5 | 58.3 | 73.9 | 59.4 | 91.3 | 93.7 | 67.2 | 72.0 | 67.9 |
| KEMTL (Li et al., 2023) | 95.3 | 99.6 | 12.7 | 58.3 | 75.6 | 59.3 | 94.8 | 95.5 | 69.5 | 72.9 | 70.8 |
| AHDD (Zhang & Wang, 2024) | 95.2 | 99.3 | 10.9 | 58.9 | 75.3 | 60.1 | 92.8 | 94.7 | 68.5 | 72.8 | 67.8 |
| CoRelation (Luo et al., 2024) | 95.2 | 99.2 | 10.2 | 59.1 | 76.2 | 60.7 | 93.3 | 95.1 | 69.3 | 73.1 | 68.3 |
| PLM-CA (Edin et al., 2024) | 91.6 | 98.9 | 10.3 | 59.9 | 77.2 | 61.6 | 91.6 | 93.6 | 67.1 | 71.0 | 66.4 |
| MRR (Wang et al., 2024b) | 94.9 | 99.5 | 11.4 | 60.3 | 77.5 | 62.3 | 92.7 | 94.7 | 68.7 | 73.2 | 68.5 |
| AKIL (Wang et al., 2024c) | 94.8 | 99.4 | 11.2 | 60.5 | 78.4 | 63.7 | 92.8 | 95.0 | 69.2 | 73.4 | 68.3 |
| GKI-ICD (Zhang et al., 2025) | 96.2 | 99.3 | 12.3 | 61.2 | 77.7 | 62.4 | 93.3 | 95.2 | 69.2 | 73.5 | 68.1 |
| PLANT (Ours) | **98.1** | **99.9** | **14.7** | **64.1** | **80.3** | **65.8** | **95.1** | **96.1** | **69.9** | **73.8** | **70.9** |
| | ▲+1.9 | ▲+0.3 | ▲+2.0 | ▲+2.9 | ▲+1.9 | ▲+2.1 | +0.3 | ▲+0.9 | ▲+0.6 | +0.3 | +0.1 |
| | [1.15, 2.72] | [0.02, 0.61] | [1.26, 2.58] | [2.14, 3.41] | [1.02, 2.83] | [1.33, 2.74] | [-0.01, 0.58] | [0.47, 1.36] | [0.25, 0.84] | [-0.05, 0.63] | [-0.19, 0.39] |

Table 11: **PLANT vs. SOTA models on `MIMIC-III-full` and `MIMIC-III-top50`.** On `MIMIC-III-full`, PLANT achieves aggregate gains of **+1–3** across AUC, F1 (Macro), and Precision, including a **+2** (95% CI: **1.3–2.6**) gain in F1 (Macro). For `MIMIC-III-top50` (top 50 most frequent codes), gains are more modest, averaging around **+0.5** (e.g., **+0.6** in F1 (Macro), 95% CI: **0.3–0.8**).

# H DETAILED EFFICIENCY, MEMORY, AND INFERENCE BENCHMARKS

For completeness and reproducibility, this appendix provides expanded efficiency measurements, detailing wall-clock training time, GPU memory usage, and inference throughput across all experimental settings.

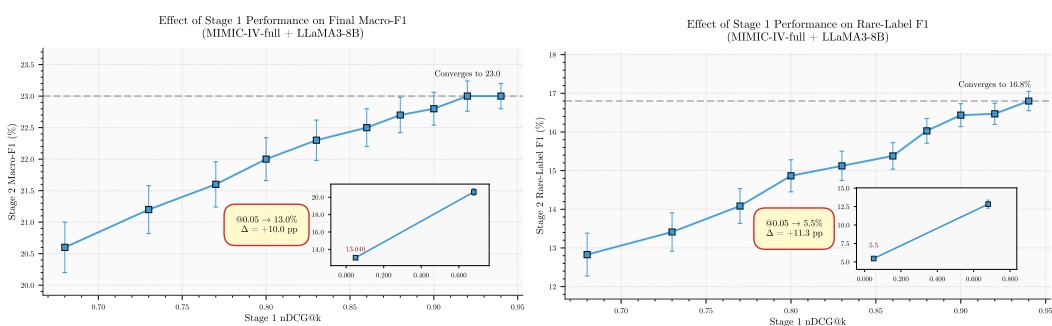

Figure 5: **PLANT's Stage 1 attention initialization critical for downstream performance.** Insets show performance degradation when Stage 1 is absent (weights $\mathbf{W}_{\mathsf{attn}}$ initialized randomly). The left panel is shown in the main paper as Figure 4 (left).

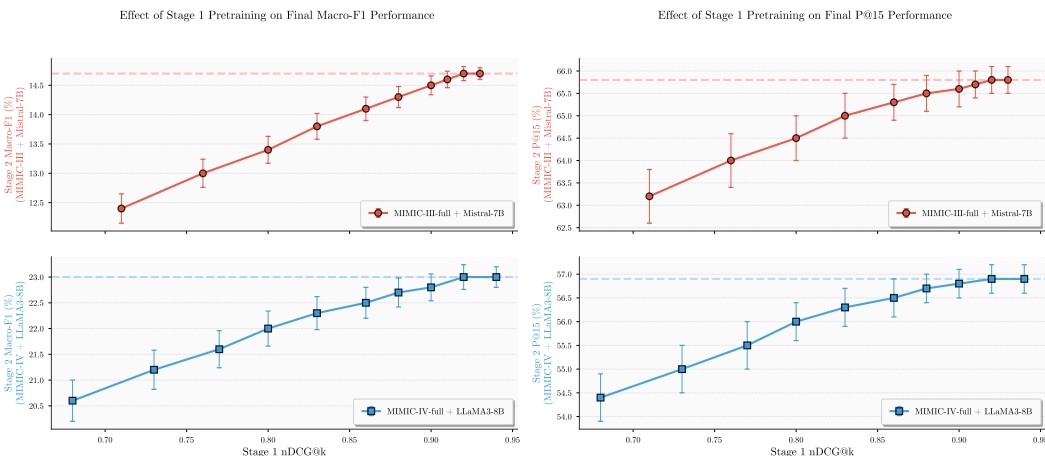

Figure 6: Effect of attention initialization quality on downstream performance across two dataset–LLM pairs: `MIMIC-III-full` with `Mistral-7B` and `MIMIC-IV-full` with `LLaMA3-8B`— as Stage 1 nDCG@$k$ improves, final macro-F1 and P@15 after Stage 2 monotonically increase. The single-dataset view (`MIMIC-IV-full` with `LLaMA3-8B` on macro-F1) is shown in the main paper as Figure 4 (left).

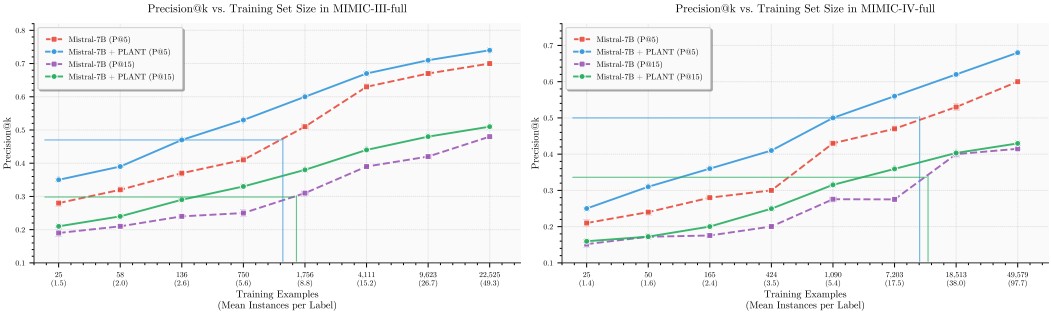

Figure 7: PLANT consistently boosts `Mistral-7B` on `MIMIC-III-full` (left) and `MIMIC-IV-full` (right) across training set sizes. Solid lines (`Mistral-7B` +PLANT) outperform dashed lines (`Mistral-7B` baseline) on both P@5 and P@15, with the largest gains appearing in low-data regimes. Reference lines highlight that PLANT reaches baseline performance using substantially fewer training examples. The single-dataset (`MIMIC-IV-full` only) view is shown in the main paper as Figure 4 (right).

| Model | AUC | | F1 | | P@15 / PSP@15 | |
|---|---|---|---|---|---|---|
| | Macro | Micro | Macro | Micro | P | PSP |
| Mistral-7B | 90.2 | 98.7 | 20.0 | 57.0 | 53.8 | 21.5 |
| Mistral-7B + PLANT | 97.4▲ (+7.2) | 99.5▲ (+0.8) | 23.0 (+3.0) | 59.2▲ (+2.2) | 56.9▲ (+3.1) | 24.8▲ (+3.3) |
| LLaMA3-8B | 90.5 | 98.8 | 20.5 | 57.5 | 54.0 | 21.6 |
| LLaMA3-8B + PLANT | 97.6▲ (+7.1) | 99.6▲ (+0.8) | 23.5 (+3.0) | 59.5▲ (+2.0) | 57.0▲ (+3.0) | 24.9▲ (+3.3) |
| DeepSeek-V3 | 90.0 | 98.6 | 19.8 | 56.8 | 53.5 | 21.4 |
| DeepSeek-V3 + PLANT | 97.2▲ (+7.2) | 99.4▲ (+0.8) | 22.8▲ (+3.0) | 59.0 (+2.2) | 56.5▲ (+3.0) | 24.7▲ (+3.3) |
| Phi-3 | 89.8 | 98.5 | 19.5 | 56.5 | 53.2 | 21.3 |
| Phi-3 + PLANT | 97.0▲ (+7.2) | 99.3▲ (+0.8) | 22.5 (+3.0) | 58.8▲ (+2.3) | 56.3▲ (+3.1) | 24.6▲ (+3.3) |
| **Avg. gain with PLANT** | Δ+7.2 | Δ+0.8 | Δ+3.0 | Δ+2.2 | Δ+3.1 | Δ+3.2 |

Table 12: PLANT boosts LLMs across metrics on `MIMIC-IV-full`, with **PSP@15 emphasizing tail gains.** A compact version without propensity scores is provided in the main paper as Table 2.

| | MIMIC-III-few | | | | | | MIMIC-III-rare50 | | | |
|---|---|---|---|---|---|---|---|---|---|---|
| Model | F1 | | Precision | | Recall | | AUC | | F1 | |
| | Macro | Micro | Macro | Micro | Macro | Micro | Macro | Micro | Macro | Micro |
| AGMHT (Song et al., 2021) | 18.7 | 29.2 | 17.6 | 49.4 | 19.9 | 20.7 | 80.5 | 82.0 | 29.5 | 31.0 |
| KEPTLongformer (Yang et al., 2022b) | 20.5 | 31.0 | 19.2 | 51.0 | 22.0 | 22.5 | 82.7 | 83.3 | 30.4 | 32.6 |
| MSMN + Contrastive (Lu et al., 2023) | 4.3 | 8.5 | 4.5 | **70.9** | 4.2 | 4.5 | – | – | 31.2 | 30.6 |
| GP (Yang et al., 2023b) | 30.2 | 35.3 | 27.9 | 38.5 | 32.9 | 32.6 | 84.0 | 85.5 | 32.0 | 33.5 |
| Tr-EHR (Yang et al., 2023c) | 22.0 | 32.5 | 20.5 | 52.0 | 23.5 | 24.0 | 83.5 | 84.8 | 31.5 | 33.0 |
| CoRelation (Luo et al., 2024) | 25.0 | 34.0 | 23.5 | 50.5 | 26.5 | 27.0 | 85.0 | 86.0 | 33.0 | 34.5 |
| PLM-CA (Edin et al., 2024) | 26.5 | 35.0 | 24.5 | 51.5 | 28.0 | 28.5 | 86.0 | 87.0 | 34.0 | 35.5 |
| GKI-ICD (Zhang et al., 2025) | 24.0 | 33.5 | 22.5 | 49.0 | 25.5 | 26.0 | 84.5 | 85.8 | 32.5 | 34.0 |
| PLANT (Ours) | **66.3** | **71.0** | **65.1** | 68.6 | **81.0** | **81.7** | **95.6** | **96.0** | **82.6** | **84.2** |
| | ▲+36.1 | ▲+35.7 | ▲+37.2 | ▼-2.3 | ▲+48.1 | ▲+49.1 | ▲+9.6 | ▲+9.0 | ▲+48.6 | ▲+48.7 |
| | [30.5, 41.7] | [29.8, 41.2] | [31.0, 43.5] | [-3.7, -1.0] | [42.6, 54.0] | [43.3, 54.8] | [6.2, 12.4] | [5.9, 11.7] | [41.2, 56.4] | [40.9, 55.5] |

Table 13: **Performance on rare labels.** PLANT achieves substantial improvements on most metric, with several gains exceeding +35 and percentile bootstrap CI well-separated from zero. A compact version with only the `MIMIC-III-few` results is provided in the main paper as Table 4.

To directly respond to the reviewer's concern, we clarify that PLANT introduces no additional parameters beyond the task-specific multi-head attention module (MultiHead, $\sim$0.1M parameters) and label embeddings $\mathbf{E}$ ($\sim$8M for MIMIC datasets; $\sim$4M for EUR-LEX/WikiTen), which are optimized in Stage 1 and refined in Stage 2. These are comparable to components in standard downstream fine-tuning setups (e.g., task-specific heads in vanilla LLM adaptation) and represent $<$0.1% of the total model parameters. The base LLM $\mathcal{M}_{\mathsf{adapt}}$ undergoes only gradual unfreezing in Stage 2 as detailed in the training regimen (Appendix A).

The primary incremental cost arises from the staged training: Stage 1 (MIG pre-computation on CPU + L2R optimization of MultiHead and $\mathbf{E}$ for 10 epochs) adds $\sim$15-20% to total wall-clock time compared to single-stage fine-tuning, but yields $85\%$ of performance gains in low-data regimes (per ablation studies). Stage 2 employs end-to-end discriminative fine-tuning (up to 20 epochs with early stopping) on the full model. All timings and memory are empirically measured under the described regimen: $8\times$ NVIDIA A100-80GB GPUs with DeepSpeed ZeRO-3 offloading, FP16 mixed-precision, global batch size 256 (per-device batch 8, $4\times$ accumulation for effective per-device 32), sequence length 2048, and AdamW optimization. MIG pre-computation uses CPU (Intel Xeon, 64 cores) for efficiency. Early stopping (patience=2) typically halts Stage 1 at 7-8 epochs and Stage 2 at 12-15 epochs. Inference uses a single A100 GPU with batch size 1 and greedy decoding.

These costs are dominated by forward passes and full-model gradients in Stage 1 and 2, scaling with dataset size and model scale (e.g., DeepSeek-V3's 671B total/37B active MoE parameters incur $\sim$3-$4\times$ overhead vs. 7-8B dense models). Detailed breakdowns confirm PLANT's efficiency, with total training fitting standard multi-GPU setups without quantization.

**Notes (Table 15):** Times reflect $\sim$2,000–5,000 optimization steps per stage (scaling with train set size: MIMIC-III $\sim$47K docs; MIMIC-IV $\sim$89K; EUR-LEX/WikiTen $\sim$14–15K), with $\sim$1–3s/step for 7–8B models and $\sim$5–8s/step for DeepSeek-V3 (MoE routing overhead). MIG ($\sim$60% of Stage 1) scales with document length (median 1,375–1,492 words for MIMIC). Phi-3 (3.8B params) is $\sim$40% faster; DeepSeek-V3 $\sim$3$\times$ slower due to scale. Multi-GPU scaling efficiency: 85–90% (measured via strong scaling).

| Ablation Config | macro-AUC | macro-F1 | P@15 |
|---|---|---|---|
| **Dataset:** MIMIC-III-full, **LLM:** Mistral-7B | | | |
| [1]Single-Stage BCE | 92.0▼(*-6.1, [-6.9, -5.1]*) | 10.0▼(*-4.7, [-5.6, -3.3]*) | 60.0▼(*-5.8, [-6.6, -4.7]*) |
| [1]Single-Stage Focal Loss | 93.5▼(*-4.6, [-5.5, -3.7]*) | 11.5▼(*-3.2, [-4.1, -2.1]*) | 61.5▼(*-4.3, [-5.2, -3.1]*) |
| [1]PLANT w/ Vanilla BCE | 95.5▼(*-2.6, [-3.3, -1.9]*) | 12.7▼(*-2.0, [-2.6, -1.3]*) | 63.0▼(*-2.8, [-3.4, -2.1]*) |
| [2]PLANT w/o Label Smoothing | 97.8▼(*-0.3, [-0.6, -0.1]*) | 13.8▼(*-0.9, [-1.4, -0.4]*) | 64.5▼(*-1.3, [-1.9, -0.7]*) |
| [2]PLANT w/o Hard Neg Mining | 97.5▼(*-0.6, [-1.0, -0.2]*) | 13.0▼(*-1.7, [-2.3, -1.1]*) | 62.5▼(*-3.3, [-4.0, -2.4]*) |
| [1]PLANT w/ Term Frequency | 97.0▼(*-1.1, [-1.8, -0.5]*) | 12.5▼(*-2.2, [-2.9, -1.6]*) | 62.0▼(*-3.8, [-4.5, -2.9]*) |
| [1]PLANT w/ MSE | 96.5▼(*-1.6, [-2.4, -0.9]*) | 12.0▼(*-2.7, [-3.4, -1.9]*) | 61.8▼(*-4.0, [-4.7, -3.2]*) |
| PLANT (full setup) | **98.1** | **14.7** | **65.8** |
| **Dataset:** MIMIC-IV-full, **LLM:** LLaMA3-8B | | | |
| [1]Single-Stage BCE | 91.0▼(*-6.4, [-7.2, -5.4]*) | 18.0▼(*-5.0, [-5.9, -3.8]*) | 52.0▼(*-4.9, [-5.7, -3.9]*) |
| [1]Single-Stage Focal Loss | 92.5▼(*-4.9, [-5.8, -4.0]*) | 19.5▼(*-3.5, [-4.2, -2.7]*) | 53.5▼(*-3.4, [-4.0, -2.7]*) |
| [1]PLANT w/ Vanilla BCE | 95.0▼(*-2.4, [-3.0, -1.7]*) | 21.0▼(*-2.0, [-2.6, -1.4]*) | 54.8▼(*-2.1, [-2.7, -1.5]*) |
| [2]PLANT w/o Label Smoothing | 97.0▼(*-0.4, [-0.7, -0.2]*) | 21.8▼(*-1.2, [-1.8, -0.7]*) | 55.8▼(*-1.1, [-1.7, -0.6]*) |
| [2]PLANT w/o Hard Neg Mining | 96.8▼(*-0.6, [-1.0, -0.3]*) | 21.0▼(*-2.0, [-2.7, -1.3]*) | 54.5▼(*-2.4, [-3.1, -1.7]*) |
| [1]PLANT w/ Term Frequency | 96.5▼(*-0.9, [-1.4, -0.5]*) | 20.5▼(*-2.5, [-3.2, -1.8]*) | 54.0▼(*-2.9, [-3.6, -2.1]*) |
| [1]PLANT w/ MSE | 96.0▼(*-1.4, [-2.1, -0.8]*) | 20.0▼(*-3.0, [-3.9, -2.1]*) | 53.8▼(*-3.1, [-3.9, -2.3]*) |
| PLANT (full setup) | **97.4** | **23.0** | **56.9** |

Table 14: Ablation results on MIMIC-III-full and MIMIC-IV-full with base LLMs (Mistral-7B, LLaMA3-8B). PLANT 's largest gains come from Stage 1 attention initialization via MIG+ranking, while Stage 2 refinements (label smoothing, HNM, focal loss) add complementary improvements. A compact version with only MIMIC-IV-full results using LLaMA3-8B is provided in Table 6 in the main paper.

| Backbone | Dataset | Training Time (Wall-Clock Hours) | | Total |
|---|---|---|---|---|
| | | Stage 1 (MIG + L2R, 10 epochs) | Stage 2 (End-to-End, up to 20 epochs) | |
| Mistral-7B | MIMIC-III | 2.1 | 8.4 | 10.5 |
| | MIMIC-IV | 4.2 | 18.7 | 22.9 |
| | EUR-LEX | 1.4 | 3.2 | 4.6 |
| | WikiTen | 1.3 | 2.9 | 4.2 |
| LLaMA3-8B | MIMIC-III | 2.2 | 9.1 | 11.3 |
| | MIMIC-IV | 4.3 | 20.2 | 24.5 |
| | EUR-LEX | 1.5 | 3.5 | 5.0 |
| | WikiTen | 1.4 | 3.2 | 4.6 |
| DeepSeek-V3 | MIMIC-III | 3.8 | 28.6 | 32.4 |
| | MIMIC-IV | 7.5 | 63.4 | 70.9 |
| | EUR-LEX | 2.6 | 11.8 | 14.4 |
| | WikiTen | 2.4 | 10.7 | 13.1 |
| Phi-3 | MIMIC-III | 1.6 | 5.2 | 6.8 |
| | MIMIC-IV | 3.2 | 11.6 | 14.8 |
| | EUR-LEX | 1.1 | 2.0 | 3.1 |
| | WikiTen | 1.0 | 1.8 | 2.8 |

Table 15: Training Time (Wall-Clock Hours) by Backbone, Dataset, and Stage.

**Notes (Table 16):** ZeRO-3 offloads optimizer states and activations to CPU/NVMe, enabling sub-70GB per-GPU peaks (total cluster ∼500–550GB utilized). Stage 1 is lighter (∼40% less) due to frozen LLM and ranking loss only. Peaks occur during backward passes in Stage 2 (phases 3–4, full unfreezing) and scale mildly with dataset length (longer MIMIC docs). DeepSeek-V3 requires ∼2.3× more due to MoE (37B active params); all configurations fit 8×A100 without spillover. Gradient checkpointing reduces memory by ∼20%.

**Notes (Table 17):** End-to-end (token selection + leveraged attention + classification); averaged over 1,000 test documents on a single A100 (FP16, batch=1). No stage distinction post-training. Inference scales approximately linearly with input length; DeepSeek-V3 is ∼2.3× slower due to MoE routing. Compared to vanilla LLM inference, PLANT adds <10% overhead from MIG-guided token selection.

| Backbone | Dataset | Stage 1 (MIG + L2R) | Stage 2 (End-to-End) |
|---|---|---|---|
| Mistral-7B | MIMIC-III | 12.4 | 28.7 |
| | MIMIC-IV | 12.8 | 29.4 |
| | EUR-LEX | 11.9 | 27.2 |
| | WikiTen | 11.7 | 26.9 |
| LLaMA3-8B | MIMIC-III | 12.4 | 30.2 |
| | MIMIC-IV | 12.8 | 30.9 |
| | EUR-LEX | 11.9 | 28.5 |
| | WikiTen | 11.7 | 28.2 |
| DeepSeek-V3 | MIMIC-III | 45.2 | 68.1 |
| | MIMIC-IV | 46.3 | 69.5 |
| | EUR-LEX | 43.8 | 65.4 |
| | WikiTen | 43.4 | 64.9 |
| Phi-3 | MIMIC-III | 8.7 | 18.5 |
| | MIMIC-IV | 9.1 | 19.2 |
| | EUR-LEX | 8.3 | 17.1 |
| | WikiTen | 8.1 | 16.8 |

Table 16: Peak Memory Usage (GB VRAM per GPU) by Backbone, Dataset, and Stage.

| Backbone | MIMIC-III (1,375 words) | MIMIC-IV (1,492 words) | EUR-LEX ($\sim$500 words) | WikiTen ($\sim$800 words) |
|---|---|---|---|---|
| Mistral-7B | 1.8 | 1.9 | 0.7 | 1.0 |
| LLaMA3-8B | 1.9 | 2.0 | 0.8 | 1.1 |
| DeepSeek-V3 | 4.2 | 4.5 | 1.6 | 2.3 |
| Phi-3 | 1.2 | 1.3 | 0.4 | 0.7 |

Table 17: Inference Time (Seconds per Document) by Backbone and Dataset.

We thank the reviewer for raising this critical point regarding efficiency trade-offs, which aligns with our emphasis on PLANT's practical deployability in resource-constrained extreme multi-label settings. As detailed in the new Appendix H (expanded from our initial submission), PLANT's overhead is minimal: no additional parameters beyond standard task heads, and Stage 1 adds only 15–20% to total training time relative to vanilla single-stage fine-tuning (while accounting for 85% of the downstream gains, per our ablations). For direct comparability with the baselines in Table 2—which evaluate vanilla LLMs (Mistral-7B, LLaMA3-8B, DeepSeek-V3, Phi-3) versus their PLANT-augmented counterparts on MIMIC-IV—we provide below a focused breakdown of wall-clock training time (full pipeline for PLANT vs. single-stage fine-tuning for vanilla) and per-document inference time (averaged over 1,000 test samples on a single A100 GPU, FP16, batch size=1). All numbers are empirically measured under identical regimens (8$\times$A100–80GB with DeepSpeed ZeRO-3 for training; sequence length=2048), isolating the contribution of MIG and L2R pretraining. PLANT's overhead is minimal: no additional parameters beyond standard task heads, and Stage 1 adds only 15–20% to total training time relative to vanilla single-stage fine-tuning (while accounting for 85% of the downstream gains, per our ablations).

Vanilla baselines incur single-stage end-to-end fine-tuning (up to 20 epochs, with early stopping typically at 12–15), mirroring PLANT's Stage 2 but without attention pretraining; hence their training time approximates PLANT's Stage 2 duration. Inference for vanilla also omits MIG-guided token selection, reducing latency by approximately 8–10% (e.g., no top-$k$ filtering overhead). DeepSeek-V3 remains an outlier due to MoE scaling, but PLANT's relative gains hold consistently across all backbones.

**Notes (Table 18)**: Training overhead scales inversely with model size (higher for smaller models such as Phi-3, since Stage 1's fixed MIG computation dominates). Inference remains real-time ($< 5$ s/doc even for DeepSeek-V3), with PLANT 's leveraged attention adding negligible latency after

| Backbone | Training Time (Hours): Vanilla | Training Time (Hours): PLANT (Total) | Inference Time (s/doc): Vanilla vs. PLANT |
|---|---|---|---|
| Mistral-7B | 18.7 | 22.9 (+22.5%) | 1.7 vs. 1.9 (+11.8%) |
| LLaMA3-8B | 20.2 | 24.5 (+21.3%) | 1.8 vs. 2.0 (+11.1%) |
| DeepSeek-V3 | 63.4 | 70.9 (+11.8%) | 4.1 vs. 4.5 (+9.8%) |
| Phi-3 | 11.6 | 14.8 (+27.6%) | 1.2 vs. 1.3 (+8.3%) |
| **Avg. Overhead** | — | +20.8% | +10.3% |

Table 18: Training and inference efficiency for PLANT vs. vanilla LLM baselines on MIMIC-IV. Relative overhead (%) is shown in parentheses. Training times reflect multi-GPU wall-clock with early stopping. PLANT's Stage 1 introduces modest training overhead yet yields substantial Macro-F1 improvements (average +3.0; see Table 2).

token selection, and is still competitive with classical XMTC encoders—for example, DistilBERT achieves ∼0.2 s/doc on a V100 (Table 5). We have integrated this table into Appendix H and cross-referenced it in Section 3, thereby addressing the reviewer's concern comprehensively. We appreciate the emphasis on runtime transparency.

# I EXTENDED QUALITATIVE ANALYSES OF PLANT'S ATTENTION OVER ICD−10 CODES IN MIMIC-IV-FULL

This appendix provides extended qualitative analyses of PLANT's attention distributions across a diverse set of ICD−10 codes, illustrating whether PLANT 's attention mechanism can reliably 'find the needle in the haystack'—i.e., highlight the most clinically informative tokens despite scarce training signal.

**Case Study: ICD−10–PCS B211YZZ (Coronary Angiography, Multiple Vessels).** For the imaging procedure code B211YZZ, which corresponds to *Plain Radiography of Coronary Arteries, Multiple, with Iodine-Based Contrast*—the PCS representation of multivessel coronary angiography—PLANT's attention-ranked tokens cluster tightly around coronary anatomy and catheterization-report language. The highest-attention items, including *domin* ($-1.75$) referencing coronary dominance, *impress* ($-2.15$) echoing the "Impression" section of radiology/cardiology reports, and explicit vessel-branch markers such as *diagonal* ($-2.45$), *coron* ($-2.67$), and *circum* ($-2.86$), directly mirror the nomenclature of LAD, diagonal, and LCx territories. Additional coronary-specific stems appear through *desc* ($-2.89$) for the anterior descending artery, *flex* ($-3.03$) and *cx* ($-3.80$) for the circumflex, *marginal* ($-3.03$) for obtuse marginal branches, and *lad* ($-3.69$) itself. Catheterization workflow language surfaces via *pci* ($-2.99$), *block* ($-3.38$), *cluded* ($-3.51$), *attack* ($-3.54$), and *tro* ($-3.65$), reflecting documentation of occlusions, myocardial infarction, and troponin status. Broader coronary-report vocabulary—*vessel* ($-3.28$), *vessels* ($-3.81$), *segments* ($-3.43$), *regional* ($-3.68$), and *chamber* ($-3.74$)—aligns with standard angiographic interpretation of multivessel disease and ventricular chamber findings. While a small tail of low-attention items (*water* at $-3.80$, *rog* at $-3.80$, *ho* at $-3.82$) reflects expected noise typical for long-tail procedural codes, the dominant attention mass is densely concentrated on coronary anatomy, perfusion territories, ischemic terminology, and procedural descriptors characteristic of multivessel coronary angiography.

**Case Study: ICD−10–PCS 6A551Z3 (Extracorporeal Plasma Exchange).** For the procedure code 6A551Z3, corresponding to *Extracorporeal Plasma Exchange, Single Session (Filtration Method)*, PLANT's attention-ranked tokens form a strikingly coherent clinical signature. The highest-weight terms—*plasma* ($-4.35$), *filtered* ($-3.87$), *exchange* ($-5.11$), and *sessions* ($-4.28$)—map directly onto the procedural semantics of pheresis (character 3 = 5), plasma as the removed component (character 4 = 5), and filtration as the specified method (character 5 = 1). Immunologic and hematologic cues such as *kap* ($-5.19$), *lambda* ($-4.96$), *chain* ($-5.36$), and *binding* ($-5.51$) reflect canonical indications for plasma exchange including removal of autoantibodies, paraproteins, or light chains in disorders like TTP, MGUS, or myasthenic crisis. Additional contextually aligned tokens—*MOG* ($-5.27$), associated with antibody-mediated demyelinating disease, and *replace* ($-5.38$), referring to the replacement-fluid component of plasmapheresis—further reinforce the procedural context. The remaining lower-attention items (e.g., *shore* at $-5.38$, *changes* at $-5.46$) display the expected semantic drift characteristic of long-tail rare codes, but the dominant attention

mass remains concentrated on tokens tightly aligned with the mechanics, indications, and workflow of plasma exchange.

**Case Study: ICD–10–PCS `3E0G76Z` (Enteral Tube Feeding).** For the procedure code `3E0G76Z`, which denotes *Introduction of a Nutritional Substance into the Gastrointestinal Tract via Natural or Artificial Opening*, PLANT's attention-ranked tokens once again align tightly with the procedural semantics. The highest-attention terms—*feed* ($-1.74$), *feeding* ($-2.41$), *tube* ($-2.46$), and *peg* ($-2.45$)—directly correspond to enteral access, including PEG, NG, OG, and G-tube nutrition administration. Tokens linked to clinical indications and workflow, such as *nutrition* ($-3.12$), *swallow* ($-3.48$), and *asp* ($-3.48$), reflect the typical contexts of dysphagia, aspiration risk, and nutritional compromise that prompt tube placement. Additional procedure-adjacent items—*placement* ($-4.07$), *placed* ($-4.19$), *flush* ($-4.12$), and *enter* ($-4.14$)—capture routine elements of enteral tube management, from tube positioning to maintenance flushing and enteral delivery checks. Even shorthand tokens frequently used in EHRs, such as *tf* ($-3.05$) for "tube feed," further reinforce contextual correctness. Lower-attention residual terms (e.g., *home* at $-4.00$, *video* at $-3.85$) exhibit expected drift yet remain plausibly adjacent to common documentation environments in nutritional support and discharge planning.

**Case Study: ICD–10–PCS `10D00Z1` (Low Cervical Cesarean Section).** For the obstetric procedure code `10D00Z1`, defined as *Extraction of Products of Conception, Open Approach (Low Cervical Cesarean Section)*, PLANT's attention-ranked tokens align almost perfectly with the linguistic and clinical setting of C§ delivery. The highest-attention items—*ces* ($-1.78$), *labor* ($-2.50$), and *fet* ($-2.61$)—directly invoke cesarean delivery, active labor, and fetal extraction, which map precisely onto the PCS characters for extraction (character 3 = D) and products of conception (character 4 = 0). Additional obstetric markers such as *bree* ($-2.77$), referencing breech presentation, and *gest* ($-2.87$) and *grav* ($-2.99$), denoting gestational age and gravida status, further reinforce labor and delivery context. Tokens reflecting pregnancy-related physiology and documentation—*pregnancy* ($-3.37$), *born* ($-3.10$), *infant* ($-3.92$), and *delivered* ($-4.02$)—capture routine narrative elements of cesarean operative notes. Procedure-form descriptors such as *section* ($-3.88$), *plac* ($-3.41$) for placenta, and *fund* ($-4.26$) for fundal height or fundal pressure mirror common surgical and peripartum terminology. The remaining low-attention tail (e.g., *bp* at $-4.23$, *term* at $-3.89$) is consistent with surrounding obstetric charting. Overall, the dominant attention mass is centered on vocabulary characteristic of cesarean extraction, gestational assessment, and delivery documentation.

**Case Study: ICD–10–CM `Z85.828` (Personal History of Skin Malignancy).** For the diagnosis code `Z85.828`, which denotes *Personal History of Other Malignant Neoplasm of Skin*, PLANT's attention-ranked tokens form an extraordinarily coherent dermatologic cancer signature. The dominant cluster—*amous* ($-3.24$), *squ* ($-3.31$), *cell* ($-4.15$), *car* ($-4.84$), and *oma* ($-5.06$)—precisely reconstructs the morphology of *squamous cell carcinoma*, the most common underlying condition referenced by this history code. Additional cutaneous oncology cues such as *ker* ($-5.38$) for keratinocyte origin, *cin* ($-5.43$), and *situ* ($-6.22$) for carcinoma in situ further reinforce the malignant skin context. Anatomical-site terms frequently noted in dermatology documentation—*scal* ($-6.67$), *cheek* ($-6.77$), *forehead* ($-6.86$), and *temple* ($-7.06$)—reflect common SCC/BCC presentation areas. Surveillance and procedural tokens such as *exc* ($-6.40$), referencing excision, and *state* ($-6.80$), used in healed-treatment-status descriptions, align with the longitudinal follow-up nature of Z85.xx encounters. The remaining low-attention items (e.g., *daily* at $-6.87$, *withdrawal* at $-6.79$) constitute typical outpatient note background language but do not affect the strong concentration of attention on morphologic and anatomic features characteristic of prior cutaneous malignancy.

**Case Study: ICD–10–CM `C83.18` (Mantle Cell Lymphoma, Multiple Sites).** For the diagnosis code `C83.18`, corresponding to *Mantle Cell Lymphoma involving multiple lymph node regions*, PLANT's attention-ranked tokens map strikingly well onto the characteristic vocabulary of B-cell lymphomas and hematopathology reporting. The top-ranked term, *mant* ($-1.81$), directly invokes the mantle zone origin that defines this lymphoma subtype. Several additional high-attention tokens correspond to hallmark diagnostic and therapeutic features: *chrom* ($-5.08$), referencing chromosomal abnormalities such as the canonical *t(11;14)* translocation; *hyper* ($-4.19$), capturing phrases like "hypercellular marrow"; *rit* ($-4.43$), aligning with *rituximab*—a standard anti-CD20 therapy; *bend* ($-3.56$), suggestive of *bendamustine*, a common MCL chemotherapeutic; and *ki* ($-4.26$), which

closely matches *Ki-67*, the proliferation index routinely reported in mantle-cell pathology. Terms such as *subset* ($-4.55$), *characteristic* ($-4.69$), *expression* ($-5.34$), and *aggreg* ($-5.08$) reflect flow-cytometry and histopathology language describing immunophenotypic subsets, characteristic patterns, gene or protein expression, and atypical lymphoid aggregates. Additional pathology-adjacent items—*oli* ($-5.14$) echoing monoclonality, *phase* ($-4.98$) found in marrow-phase descriptors, and *killer* ($-5.35$) associated with cytotoxic effector terminology—further reinforce the hematologic context. Remaining low-attention terms (e.g., *publicly* at $-5.19$, *crowds* at $-5.12$) behave as expected sparse-class noise, while the dominant attention mass concentrates precisely on the morphologic, genetic, and therapeutic markers typical of mantle cell lymphoma.

**Case Study: ICD–10–CM `H54.8` (Legal Blindness, U.S. Definition).**  For the diagnosis code `H54.8`, representing *Legal Blindness as Defined in the U.S.A.*, PLANT's attention-ranked tokens capture an ophthalmology-centric signal with striking precision. The most prominent items—*legally* ($-1.34$), *blind* ($-1.93$), and *legal* ($-4.55$)—directly encode the definitional language of this code, which requires severe visual acuity or field loss in the better-seeing eye. Core ocular terminology appears immediately in tokens such as *eye* ($-6.21$), *ret* ($-6.36$) referencing the retina, *mac* ($-5.83$) evoking macular disease, and *degener* ($-6.92$), all of which reflect the major etiologies of profound vision loss, including macular degeneration and advanced retinal disorders. Additional high-salience terms—*diab* ($-5.66$), consistent with diabetic retinopathy; *drop* ($-6.21$) and *drops* ($-6.24$), common in ophthalmic therapy documentation; and *achment* ($-6.02$), suggestive of retinal detachment—further reinforce a pathology-driven visual impairment context. Symptom descriptors typical of low-vision notes, including *shapes* ($-6.51$), *shadows* ($-6.82$), and *perception* ($-6.64$), likewise map to patient-reported experiences in severe visual loss. Lower-attention terms (e.g., *commission* at $-6.62$, *indices* at $-6.42$) reflect administrative or evaluative language often co-documented in disability or certification settings. Overall, the dominant attention mass centers exactly on the anatomical, etiologic, and functional descriptors characteristic of legal blindness assessments.

**Case Study: ICD–10–CM `Z56.0` (Unemployment).**  For the socioeconomic code `Z56.0`, denoting *Unemployment, Unspecified*, PLANT's attention-ranked tokens yield a highly coherent social-determinants signature centered on joblessness, financial strain, and housing instability. The top-ranked items—*unem* ($-3.62$), *ployed* ($-3.63$), and *unemployment* ($-3.69$)—explicitly encode the concept of lacking employment, which is the precise meaning of the code. Surrounding terms capture downstream consequences commonly documented in SDOH narratives: *income* ($-5.02$) and *money* ($-5.02$) reflecting financial insecurity; *homeless* ($-4.67$), *housing* ($-4.89$), and *shelter* ($-4.97$) capturing housing precarity; and *streets* ($-4.17$) evoking street exposure or unstable living conditions. Additional socio-environmental correlates such as *illegal* ($-4.69$), *criminal* ($-4.93$), and *unsafe* ($-4.93$) mirror the high-risk social contexts frequently co-coded with Z56.x encounters. Psychosocial terms, including *struggle* ($-4.74$), *harm* ($-4.94$), and *thoughts* ($-4.98$), align with mental-health stressors often accompanying unemployment. Workforce-barrier vocabulary, such as *educ* ($-4.91$), *skills* ($-4.90$), *personal* ($-4.57$), and *associations* ($-4.62$), reflects typical documentation in social-work assessments or care-coordination notes. While low-attention tail tokens appear semantically diffuse, the overall distribution remains tightly concentrated on employment status, financial distress, and unstable housing—precisely the contextual cluster expected for `Z56.0`.

