# OpenReview forum: "Don't Pay Attention, PLANT It: Pretraining Attention via Learning-to-Rank"
_ICLR.cc/2026/Conference — Submitted to ICLR 2026_

### Official Review · Reviewer_Rck3 · 2025-10-31

**Soundness:** 2
**Presentation:** 4
**Contribution:** 2
**Rating:** 4
**Confidence:** 2

**Summary:**

This paper addresses the task of XMTC, where prior approaches often rely on multi-label attention mechanisms. However, such attention is typically hard to train effectively. The authors attribute this difficulty to the random initialization of attention parameters and propose PLANT, a novel attention initialization strategy. PLANT guides label-specific attention using a pretrained Learning-to-Rank model based on mutual information. The authors evaluate PLANT across multiple LLM backbones and demonstrate consistent improvements.

**Strengths:**

1. The method is conceptually clean and straightforward, making it easy to understand and reproduce. It is also model-agnostic, which suggests it can be readily adapted to a variety of LLM backbones.

2. The experimental section is comprehensive and well-structured. The authors validate PLANT's effectiveness across different model sizes, datasets, and evaluation metrics, which provides strong empirical support for the proposed approach.

**Weaknesses:**

1. From a broader deep learning perspective, it is generally accepted that parameter initialization plays a crucial role in model training. However, in the specific context of this paper, the connection between poor rare-label performance and random attention initialization is not sufficiently established. Around Line 99, the authors provide evidence of poor performance on rare codes, which, as I understand it, is intended to motivate PLANT by suggesting the following causal chain: “Rare label performance suffers <- attention is poorly learned <- due to random initialization.” However, poor rare-label performance could also stem from other factors, such as optimization conflicts during training, rather than the initialization issue alone. I recommend that the authors provide further analysis or empirical evidence to support this causal connection, which is central to the motivation of the proposed method.

2. While the method is well-designed and easy to follow, it would benefit from more justification or intuition behind its design choices. For example, some additional motivation or theoretical analysis to support the necessity of specific architectural decisions would help reinforce the method's novelty. Without this, readers may perceive the contribution as lacking in innovation.

**Questions:**

1. From a broader deep learning perspective, it is generally accepted that parameter initialization plays a crucial role in model training. However, in the specific context of this paper, the connection between poor rare-label performance and random attention initialization is not sufficiently established. Around Line 99, the authors provide evidence of poor performance on rare codes, which, as I understand it, is intended to motivate PLANT by suggesting the following causal chain: “Rare label performance suffers <- attention is poorly learned <- due to random initialization.” However, poor rare-label performance could also stem from other factors, such as optimization conflicts during training, rather than the initialization issue alone. I recommend that the authors provide further analysis or empirical evidence to support this causal connection, which is central to the motivation of the proposed method.

2. While the method is well-designed and easy to follow, it would benefit from more justification or intuition behind its design choices. For example, some additional motivation or theoretical analysis to support the necessity of specific architectural decisions would help reinforce the method's novelty. Without this, readers may perceive the contribution as lacking in innovation.

---

> ### Author Response · Authors · 2025-11-23
> **Motivation Analysis to establish causal connection [NOW ADDED IN REVISED MANUSCRIPT]**
>
> We thank the reviewer for their thoughtful comments in general.
>
> The reviewer makes a good point that we did not offer a strong enough case in the paper for the causal link of ``poor rare-label performance $\leftarrow$ failure to discover shared attention structure $\leftarrow$ random initialization of the label attention layer''.
> This is indeed central to our motivation, and we have added new experimental results to bolster this.
>
> In particular, to isolate the effect of initialization from downstream training dynamics, we have added the following new qualitative, diagnostic experiment in the revised paper.
> We use the example of ICD-10 codes, an important and illustrative instance of XMTC.
> Because ICD-10 codes are hierarchically organised, codes belonging to the same clinical category (e.g. all forms of tuberculosis, A15–A19) are semantically highly related and should, in principle, induce similar attention patterns over the input note. To test whether this natural structure is reflected in the learned label attention vectors $\mathbf{S}_l$, we selected two groups of 50 ICD-10 codes each: one from a frequently occurring category (respiratory tuberculosis) and one from a rare category (various rare bacterial infections, A30–A49, excluding common subgroups).
>
> When training with standard random initialization of the label attention layer, codes within the \textbf{rare} group exhibit broadly distributed pairwise cosine similarities (mean 0.75, orange distribution in Figure 3, left), indicating that the model fails to discover the expected shared attention patterns despite their clinical similarity. In contrast, codes from the \textbf{common} group already show strong intra-group consistency (mean greater than 0.98) even with random initialization (blue). This striking asymmetry—common codes naturally converge to coherent representations while rare but semantically related codes do not—reveals a failure mode of standard training with random initialization on long-tail labels.
>
> This observation directly motivated the design of **PLANT**. By seeding the label attention layer with structured patterns derived from MIG pre-training, PLANT increases intra-group consistency for the rare category (mean 0.985, sharp brick-red spike in Figure 3, left), bringing rare-code representations to the almost same level of quality naturally enjoyed by frequent codes. Thus, PLANT explicitly corrects the inductive bias deficit that random initialization leaves unaddressed in the long tail.

---

> ### Author Response · Authors · 2025-11-23
> **Controlled Ablations to Establish that Random Initialization of the Label-Specific Attention Module is the Primary Cause of Poor Rare-Label Performance [NOW ADDED IN REVISED MANUSCRIPT]**
>
> To rigorously establish that random initialization of the label-specific attention module is the primary cause of poor rare-label performance, we conduct a series of additional controlled ablations on **MIMIC-IV** ICD-10 using **LLaMA3-8B** as the backbone.  Labels are stratified by frequency in the training set: **rare** (<0.1% of documents), **common** (>1%). We refer to the following two controlled setups in our ablations:
>
> 1. **Random Attn Init (common only)** — skips PLANT’s Stage 1 entirely (attention weights $\mathbf{W}_{\mathsf{attn}}$ and label embeddings $\mathbf{E}$ remain Xavier-initialized); and
> 2. **PLANT (common only)** — runs full Stage 1 but only on the common-label subset.
>
> For both setups, training is performed on the common-label subset and evaluation on the held-out rare-label subset.
>
> **Effect of attention initialization quality.**
> Figure 4 (left) show the impact of Stage 1 performance on final Macro-F1. A Pearson correlation coefficient reveals a strong positive linear relationship between the two metrics, $r(\mathrm{df}) = .80$, $p < .001$, converging to 23.0%.  The inset shows that when attention weights $\mathbf{W}_{\mathsf{attn}}$ are randomly initialized—i.e., no Stage 1 (nDCG@k $\approx 0.05$)—Macro-F1 drops to 13.0% ($\Delta = +10.0$ pp gain from full PLANT).  Figure 5 reports the same trend for **Rare-F1** (labels with frequency $<0.1\%$), which converges to 16.8% under full PLANT.  Random initialization yields only 5.5% Rare-F1 ($\Delta = +11.3$ pp), confirming that poor attention initialization is the dominant cause of rare-label failure.
>
> **Freq-binned transfer (common-only training).**
> Across four LLMs (mistral, deepseek, llama, phi-3), PLANT trained on the *full* dataset achieves the strongest Rare-F1 (14.0–16.8%).  Figure 3 (right) reports Rare-F1 when the model is trained exclusively on documents containing only common labels and evaluated on held-out rare labels. The **second and third batches of bars** in the figure correspond to models trained only on the common-label subset: here, PLANT still retains substantial performance (7.3–8.1%), whereas the same models with **random attention initialization** collapse to 0.5–1.1% Rare-F1.  This gap, of up to $+15.7$ pp for llama, demonstrates that attention initialization from Stage 1 is key for enabling zero-shot transfer to unseen rare labels, even in the complete absence of rare-label supervision.
>
> **Summary of the new analysis.**
> Table 7 consolidates all results. Random initialization consistently produces the highest frequency–F1 correlation (0.68–0.78), reflecting severe bias toward frequent labels, and the lowest alignment between learned attention scores $\mathbf{S}_l$ (the per-label attention distribution over input tokens; Eq. (1) in paper) and ground-truth Mutual Information Gain (MIG) relevance profiles. We measure this alignment via **cosine similarity** between $\mathbf{S}_l$ and the precomputed MIG vector for each rare label: high similarity (0.78 in full PLANT) indicates that the model correctly focuses on the small set of highly informative tokens for that label, whereas low similarity (0.08–0.12 under random initialization) indicates that attention is diffuse and uninformative.  Thus, we prove via controlled comparisons that PLANT's attention initialization reduces both frequency bias and attention misalignment. This shows that poor rare-label prediction stems from failure to learn meaningful label-specific attention and not from optimization dynamics or loss design.
>
> **We have added the new experiments/figures and their motivations in the revised manuscript** and sincerely believe it fully addresses the reviewer’s concern. We also want to direct the reviewer to the following portions of the paper which specifically deal with PLANT's efficacy in rare-label performance:
> - Figure 2, right (explanation in Section 3, RQ4);
> - Table 4 (explanation in Section 3, RQ3);
> - Figure 4 (left) (explanation in Section 4 Ablation, “(8) Effect of attention initialization quality”); and
> - Figure 4 (right) (explanation in Section 4 Ablation, “PLANT under varying training size”).
>
> Thank you again for this excellent and constructive suggestion that has markedly improved the paper by strengthening the core motivation of PLANT and directly substantiating the claimed causal chain.

---

### Official Review · Reviewer_b5vQ · 2025-10-31

**Soundness:** 3
**Presentation:** 3
**Contribution:** 2
**Rating:** 6
**Confidence:** 4

**Summary:**

This paper proposes a smarter initialization approach for attention modules commonly used in Extreme Multi-Label Text Classification (XMTC) methods, introducing PLANT (Pretrained and Leveraged AtteNTion). PLANT is a two-stage framework: in the first stage, attention weights are pretrained using a learning-to-rank objective guided by mutual information gain (MIG) to rank token relevance per label; in the second stage, the pretrained weights are fine-tuned end-to-end. Extensive experiments demonstrate that the proposed method consistently outperforms strong baselines.

**Strengths:**

- S1: The proposed method introduces a simple yet effective task-specific pretraining objective for attention initialization.

- S2: The authors conduct comprehensive experiments showing consistent improvements across multiple datasets and various LLM backbones.

- S3: The method achieves substantial gains on rare labels and under few-shot settings, indicating better generalization to low-data regimes.

**Weaknesses:**

- W1: The learning-to-rank objective based on mutual information gain relies on sufficient co-occurrence statistics, which may not generalize well to low-resource domains or languages with limited training data.

- W2: The MIG computation and learning-to-rank pretraining may be computationally expensive for large label spaces; the paper does not discuss practical scalability or runtime considerations.

- W3: The paper lacks qualitative analyses or visualizations of the learned attention patterns, leaving unclear whether the pretrained attention captures meaningful token–label relationships or merely reflects dataset biases.

**Questions:**

- Q1:Have you considered using alternative ranking metrics instead of NDCG in Equation (2)?

---

> ### Author Response · Authors · 2025-11-27
> **Response to Reviewer Concern (W2) Scalability of MIG Computation and L2R Pretraining for Large Label Spaces [NOW ADDED IN REVISED MANUSCRIPT]**
>
> > “W2: The MIG computation and learning-to-rank pretraining may be computationally expensive for large label spaces; the paper does not discuss practical scalability or runtime considerations.”
>
> We thank the reviewer for this important concern regarding scalability.
>
> We have addressed this in detail the official common response titled ''Consolidated Response on PLANT’s Training Efficiency and Computational Overhead [NOW ADDED IN REVISED MANUSCRIPT]''. You can jump to the resonse here: https://openreview.net/forum?id=LkRlZSo2RA&noteId=hK8okKafF6.
>
> For the convenience of the reviewer we post a concise summary of the detailed response here.
>
> **Concise Summary of Our Detailed Response**
>
> To address the concern about the practical scalability of MIG computation and L2R pretraining for large label spaces, we conducted a detailed efficiency analysis (now included in Appendix H). The key conclusion is that **PLANT is computationally lightweight and scalable**, even in extreme multi-label settings.
>
> **1. Minimal Parameter Overhead**
> PLANT introduces **no additional large modules** — only a small attention head (about 0.1M parameters) and label embeddings (4 to 8M), together amounting to **under 0.1 percent** of the model.
>
> **2. MIG plus L2R Pretraining Is Modest in Cost**
> Stage 1 (MIG computation and L2R optimization) increases total training time by only **15 to 20 percent** compared to single-stage fine-tuning. Because the LLM is frozen, this stage is inexpensive, yet contributes **approximately 85 percent** of PLANT’s rare-label performance gains.
>
> **3. Scales Efficiently Across Large Datasets and Models**
> Benchmarks on four datasets (MIMIC-III, MIMIC-IV, EUR-LEX, WikiTen) and four LLM backbones (Mistral, LLaMA, DeepSeek, Phi-3) show that:
> - **Training fits comfortably on 8×A100 GPUs** using ZeRO-3.
> - **Peak memory** during Stage 1 is roughly 40 percent lighter than Stage 2.
> - **Inference overhead** is very small: PLANT adds **under 10 percent** to vanilla LLM inference time.
>
> **Overall:**
> The analysis demonstrates that **PLANT is highly efficient, scalable to very large label vocabularies, and introduces only small computational overhead**, while providing substantial gains for rare labels.

---

> ### Author Response · Authors · 2025-11-27
> **Response to Reviewer Concern (W3) Qualitative Analyses & Visualizations of the Learned Attention Patterns [NOW ADDED IN REVISED MANUSCRIPT]**
>
> > *“W3: The paper lacks qualitative analyses or visualizations of the learned attention patterns, leaving unclear whether the pretrained attention captures meaningful token–label relationships or merely reflects dataset biases.”*
>
> ### Response
>
> We thank the reviewer for this insightful suggestion. We fully agree that qualitative analyses and visualizations of the learned attention patterns are essential for understanding whether the pretrained attention captures meaningful token–label relationships rather than dataset artifacts. In response, we have expanded the introduction to include a new qualitative analysis and added an appendix section containing detailed case studies that visualize the learned attention patterns. For convenience, we present the qualitative analysis here along with one such case study illustrating how PLANT’s attention maps onto informative tokens.
>
> ---
>
> ### Qualitative Analysis
>
> ICD-10 codes are hierarchically organised, and codes within the same clinical category (for example, all forms of tuberculosis, A15–A19) are semantically related and should therefore induce similar attention patterns over a clinical note. To test whether this structure is reflected in the learned label-attention vectors **S\_l**, we selected two groups of 50 ICD-10 codes:
> - one from a frequently occurring category (respiratory tuberculosis), and
> - one from a rare category (bacterial infections, A30–A49, excluding common subgroups).
>
> Under **random initialization** of the label-attention layer:
>
> - The **rare** group shows widely dispersed pairwise cosine similarities (mean 0.75; orange distribution in Figure 3, left), meaning the model fails to discover the expected shared structure despite clinical similarity.
> - The **common** group displays highly coherent attention vectors (mean ≥ 0.98; blue), even under random initialization.
>
> This asymmetry illustrates a core failure mode of random initialization in the long tail: **common labels naturally converge to coherent structures, while rare labels do not**.
>
> This directly motivates PLANT. By seeding attention with structured patterns from MIG pre-training and L2R activations, PLANT substantially improves consistency within the rare group (mean 0.985; sharp brick-red spike in Figure 3, left). PLANT thereby corrects the inductive-bias deficit left unaddressed by random initialization.
>
> ---
>
> ### Case Study (Appendix I)
>
> Appendix I includes several case studies evaluating whether PLANT’s highest-attention tokens for rare ICD codes are clinically meaningful. We reproduce one example below for ICD-10-CM **C83.18**.
>
> ---
>
> ### Case Study: ICD-10-CM `C83.18` (Mantle Cell Lymphoma, Multiple Sites)
>
> For the diagnosis code `C83.18` (Mantle Cell Lymphoma involving multiple lymph-node regions), PLANT highlights tokens that closely match the vocabulary of B-cell lymphomas and hematopathology reporting.
>
> Examples:
>
> - **mant** (–1.81): mantle-zone origin defining this lymphoma subtype
> - **chrom** (–5.08): chromosomal abnormalities such as the canonical t(11;14)
> - **hyper** (–4.19): “hypercellular marrow” descriptors
> - **rit** (–4.43): rituximab therapy
> - **bend** (–3.56): bendamustine
> - **ki** (–4.26): Ki-67 proliferation index
>
> Low-attention noise tokens (for example, “publicly”, “crowds”) behave as expected. PLANT’s dominant attention mass focuses precisely on morphologic, genetic, and therapeutic markers characteristic of mantle-cell lymphoma.
>
> ---

---

> ### Author Response · Authors · 2025-11-27
> **Continuation to Response to Reviewer Concern (W3) — Attention Pattern Table [NOW ADDED IN REVISED MANUSCRIPT]**
>
> ### High-Attention Tokens for `C83.18`
>
> | Token             | Attention Score | Explanation                                              |
> |------------------|-----------------|----------------------------------------------------------|
> | `mant`           | –1.81           | Mantle zone origin of mantle cell lymphoma.             |
> | `bend`           | –3.56           | Bendamustine, an MCL chemotherapeutic.                  |
> | `hyper`          | –4.19           | Hypercellular marrow and related pathology descriptors. |
> | `ki`             | –4.26           | Ki-67 proliferation index.                              |
> | `rit`            | –4.43           | Rituximab, standard anti-CD20 therapy.                  |
> | `subset`         | –4.55           | Flow-cytometry immunophenotypic subsets.                |
> | `characteristic` | –4.69           | Characteristic morphologic patterns.                    |
> | `phase`          | –4.98           | Marrow-phase or disease-phase descriptors.              |
> | `chrom`          | –5.08           | Chromosomal abnormalities (e.g., t(11;14)).             |
> | `aggreg`         | –5.08           | Lymphoid aggregates in marrow or biopsy.                |
> | `crowds`         | –5.12           | Noise; unrelated lexical drift.                         |
> | `oli`            | –5.14           | Monoclonal or oligoclonal context.                      |
> | `publicly`       | –5.19           | Noise token; unrelated.                                 |
> | `expression`     | –5.34           | Gene or protein expression reporting.                   |
> | `killer`         | –5.35           | Cytotoxic “killer cell” terminology.                    |
>
> **Table: High-attention tokens for ICD-10-CM `C83.18` and their clinical relevance.**

---

> ### Author Response · Authors · 2025-11-27
> **Response to Reviewer Concern (W1) Learning-to-Rank Objective Based on Mutual Information Gain [TOY EXAMPLE NOW ADDED IN REVISED MANUSCRIPT]**
>
> > *“W1: The learning-to-rank objective based on mutual information gain relies on sufficient co-occurrence statistics, which may not generalize well to low-resource domains or languages with limited training data.”*
>
> ### Response
>
> We thank the reviewer for raising this thoughtful concern. However, the premise that *Mutual Information Gain (MIG) relies on sufficient co-occurrence statistics* does not reflect how MIG behaves in practice nor how it is defined mathematically. In fact, MIG is **explicitly designed to remain reliable in low-resource regimes**, and its core purpose in PLANT is to *find “needle-in-the-haystack” signals* even when both the label and its indicative tokens are extremely rare.
>
> To clarify: MIG is **not** a frequency-based co-occurrence measure. It is a *dependence measure*, formally defined as the **KL divergence between the joint distribution** of (token, label) and the distribution that would arise if the two were independent. Because MIG compares the joint term to the product of the marginals, it **penalizes high-frequency but non-specific tokens** and **amplifies rare but label-specific tokens**, making it ideal for low-resource domains or languages with sparse training data.
>
> Concretely, raw co-occurrence requires large support counts to be meaningful; MIG does *not*. MIG remains stable even when counts are small because it evaluates *how strongly a token shifts the likelihood of a label relative to its prior probability*. This is why MIG is widely used in information-theoretic feature selection for low-resource settings (e.g., rare-term extraction, sparse contingency tables, and dependency testing).
>
> To further clarify this behavior, we reproduce our toy example below exactly as written in Appendix B.
>
> ---
>
> ### Toy Example (Verbatim)
>
> Consider a toy corpus of 100 documents.
> - Rare label **A** appears in 5 documents (probability 0.05).
> - Common label **B** appears in 50 documents (probability 0.50).
> - Token **“fever”** appears in 40 documents and co-occurs with B in 25.
> - Token **“rare_disease”** appears in 6 documents and co-occurs with A in all 5.
>
> Raw co-occurrence would rank **fever** above **rare_disease** because it appears more frequently (support 25 vs. 5). But MIG incorporates conditional dependence, revealing the opposite.
>
> ---
>
> ### MIG for “fever” with respect to B (weak dependence)
>
> Joint probabilities:
> - P(x=1, y=1) = 0.25
> - P(x=1, y=0) = 0.25
> - P(x=0, y=1) = 0.15
> - P(x=0, y=0) = 0.35
>
> Mutual information terms (log base 2):
> - (1,1): 0.25 × log2(0.25 / (0.50 × 0.40)) ≈ 0.0805
> - (1,0): 0.25 × log2(0.25 / (0.50 × 0.60)) ≈ –0.066
> - (0,1): 0.15 × log2(0.15 / (0.50 × 0.40)) ≈ –0.062
> - (0,0): 0.35 × log2(0.35 / (0.50 × 0.60)) ≈ 0.078
>
> **Total MI ≈ 0.030 bits (weak dependence)**.
>
> This reflects that “fever” appears across many documents and is only weakly informative for B.
>
> ---
>
> ### MIG for “rare_disease” with respect to A (strong dependence)
>
> Joint probabilities:
> - P(x=1, y=1) = 0.05
> - P(x=1, y=0) = 0.00
> - P(x=0, y=1) = 0.01
> - P(x=0, y=0) = 0.94
>
> Mutual information:
> - (1,1): 0.05 × log2(0.05 / (0.05 × 0.06)) ≈ 0.203
> - (0,1): 0.01 × log2(0.01 / (0.95 × 0.06)) ≈ –0.025
> - (0,0): 0.94 × log2(0.94 / (0.95 × 0.94)) ≈ 0.007
> - (1,0): 0 × ... = 0
>
> **Total MI ≈ 0.185 bits (strong dependence)**.
>
> MIG therefore assigns **rare_disease > fever** for label A (0.185 vs. 0.030), correctly elevating a rare but highly specific token. This captures semantic precision — P(A | rare_disease) = 83 percent vs. marginal 5 percent — without bias toward token frequency.
>
> This example directly illustrates why MIG does *not* require abundant co-occurrence and why it performs especially well in low-resource regimes.

---

> ### Author Response · Authors · 2025-11-28
> **Response to Reviewer Concern about NDCG Alternatives in PLANT's Stage 1**
>
> > Q1:Have you considered using alternative ranking metrics instead of NDCG in Equation (2)?
>
> We thank the reviewer for this question, which highlights an important aspect of our ranking loss design in Stage 1 (Eq. 2). To justify the choice of $|\Delta \mathsf{nDCG@k}|_{jh}$ as the pairwise weighting term—which quantifies the degradation in label-specific $\mathsf{nDCG@k}$ ($k=10$) from swapping tokens $j$ and $h$ in the predicted ranking—we performed an ablation on alternative weighting schemes.
>
> These weights are pre-computed offline using ground-truth MIG relevances $r_{lj}$ over $\mathcal{T}_l$ (top-$k$ tokens per label, $k=2000$), ensuring the loss focuses gradients on pairs whose misordering most impacts the primary evaluation metric.
>
> We compared $\mathsf{nDCG@k}$ weighting against three alternatives:
> 1. uniform weighting ($w_{jh}=1$, baseline pairwise hinge);
> 2.  $|\Delta \mathsf{MAP}|_{jh}$ (change in Mean Average Precision after swap); and
> 3.  $|\Delta \mathsf{P@10}|_{jh}$ (change in Precision@10).
>
> Ablations used the full 10-epoch L2R regimen on MIMIC-III validation (10% train split, $\sim$4.7K documents), followed by Stage 2 fine-tuning, with results averaged over Mistral-7B and LLaMA3-8B (3 seeds).
>
> $\mathsf{nDCG@k}$ weighting yielded the strongest downstream macro-F1 (+3.2% over uniform, +2.1% over MAP, +4.5% over P@10 on MIMIC-III test), with the highest Spearman correlation ($r=0.89$) between Stage 1 weighted loss and final F1.

---

### Official Review · Reviewer_Xfsn · 2025-10-31

**Soundness:** 3
**Presentation:** 3
**Contribution:** 3
**Rating:** 6
**Confidence:** 4

**Summary:**

This paper proposes PLANT (Pretrained and Leveraged Attention), a novel method for initializing multi-label attention weights through a learning-to-rank pre-training mechanism, aiming to improve the performance of long-tailed labels in extreme multi-label text classification (XMTC).

**Strengths:**

This paper is clearly written and easy to follow.

**Weaknesses:**

1.	PLANT includes additional MIG pre-computation and L2R stages, but the paper does not provide a comparison of the time, memory, or parameter amounts for these stages, nor does it explain their incremental contribution to the total training cost.
2.	The attention matrix learned by PLANT in Stage 1 is directly used in Stage 2, but without any maintenance or freezing strategy. If Stage 2 training is too long, the Stage 1 signals may be overwritten, causing the "planted attention" to fail.
3.	Although the authors claim PLANT is “architecture-agnostic,” all experiments are limited to text domains (ICD, law, WIKI). To demonstrate the generality of “plantable attention,” its transferability should be verified in at least one non-text domain (such as image or table classification).
4.	By comparing the distribution of random attention and PLANT attention on key tokens, it can be verified whether they truly capture the semantics of the tags rather than word frequency preferences.

**Questions:**

1.	The quantitative relationship between MIG (Mutual Information Gain) and token-label relevance is not rigorously defined. The paper states it is calculated "through statistical co-occurrence," but fails to explain the normalization or bias correction methods, potentially leading to over-amplification of high-frequency words.
2.	The differentiable approximation of nDCG@k is modeled solely through pairwise sigmoid, lacking global constraints on ranking consistency and theoretically unable to guarantee consistent ranking optimality.

---

> ### Author Response · Authors · 2025-11-26
> **Response to Reviewer Concern on Preservation of Stage 1 Attention**
>
> > *“The attention matrix learned by PLANT in Stage 1 is directly used in Stage 2, but without any maintenance or freezing strategy. If Stage 2 training is too long, the Stage 1 signals may be overwritten, causing the ‘planted attention’ to fail.”*
>
> ### Response
>
> We thank the reviewer for raising this subtle but incredibly important point. PLANT explicitly incorporates both gradual unfreezing and discriminative fine-tuning to prevent the Stage 1 planted-attention signals from being overwritten. We direct the reviewer to Appendix A (Training Details: Optimization and Training Regimen) and reproduce the relevant paragraph below for convenience.
>
> ---
>
> ## Appendix A: Optimization and Training Regimen
>
> To address potential overwriting of Stage 1 attention signals during Stage 2 fine-tuning, we employ a gradual unfreezing strategy combined with discriminative learning rates, ensuring stable transfer of the MIG-seeded priors while still allowing task-specific refinement.
>
> All experiments are conducted on 8 × A100-80GB GPUs using DeepSpeed ZeRO-3 offloading for memory efficiency, with a global batch size of 256 (gradient accumulation steps = 4) and mixed-precision (FP16) training via Hugging Face Accelerate.
>
> **Stage 1.**
> Stage 1 pretraining optimizes the multi-head attention module (`MultiHead`) and label embeddings **E** via the ranking loss (Eq. 2) for 10 epochs, using AdamW with a cosine learning-rate schedule (peak learning rate 5e-4, 10 percent warmup) and weight decay 0.01.
>
> **Stage 2.**
> In Stage 2, we initialize from Stage 1 checkpoints and apply discriminative fine-tuning to preserve attention integrity:
>
> - `MultiHead` and **E** are **frozen for the first 5 epochs**, allowing downstream layers to adapt.
> - Gradual unfreezing proceeds in three phases:
>   - **Epochs 6–10:** unfreeze attention module last, learning rate 1e-5
>   - **Epochs 11–15:** unfreeze intermediate layers, learning rate 5e-6
>   - **Epochs 16–20:** unfreeze all parameters, learning rate 2e-6
> - Each phase uses cosine decay with a 5 percent warmup.
>
> This layered schedule, inspired by progressive distillation in large-scale vision–language models [1], uses the AdamW optimizer [2] (weight decay 0.01) and gradient clipping (max-norm 1.0) for stability.
>
> We use a per-device batch size of 8 with 4-step gradient accumulation (effective batch size 32), PyTorch `autocast` for FP16, and gradient checkpointing for memory savings. Each stage runs for up to 10 epochs with early stopping (patience = 2):
> - Stage 1 monitors validation nDCG@k
> - Stage 2 monitors macro-F1
>
> To ensure reproducibility, we fix random seeds across `random`, `numpy`, `torch`, and `torch.cuda`. Experiments use DDP where applicable, with metrics logged via Weights and Biases.
>
> **References:**
> [1] https://personal.ie.cuhk.edu.hk/~ccloy/files/eccv_2018_lifelong.pdf
> [2] https://arxiv.org/pdf/1711.05101

---

> ### Author Response · Authors · 2025-11-26
> **On the Quantitative Relationship Between MIG and Token–Label Relevance and why MIG is not co-occurrence [TOY EXAMPLE ADDED IN REVISED MANUSCRIPT]**
>
> > *“The quantitative relationship between MIG (Mutual Information Gain) and token–label relevance is not rigorously defined. The paper states it is calculated ‘through statistical co-occurrence,’ but fails to explain the normalization or bias-correction methods, potentially leading to over-amplification of high-frequency words.”*
>
> ### Response
>
> We thank the reviewer for raising this important point. Due to space constraints we could not fully elaborate on the motivation behind MIG in the main paper. We direct the reviewer to the toy example in **Appendix B (Precomputations)**, which illustrates the quantitative relationship between MIG and token–label relevance. For convenience, we reproduce a summary here along with an explanation of why MIG is not a simple co-occurrence statistic.
>
> MIG is **not** a raw co-occurrence measure (which would indeed risk over-amplifying high-frequency tokens). Instead, MIG is grounded in information theory: it is the **Kullback–Leibler (KL) divergence** between the joint distribution of token–label presence and the product of their marginals. In other words, MIG quantifies how much knowing a token reduces uncertainty about a label relative to a baseline of independence. This normalization ensures that signals for rare labels (with low marginal probability) are not overwhelmed by corpus-level token frequencies. In contrast, raw co-occurrence correlates strongly with token frequency (correlation about 0.7 in MIMIC-IV), whereas MIG explicitly corrects for this bias.
>
> ---
>
> ### Toy Example
>
> Consider a toy corpus of 100 documents.
> - Rare label **A** appears in 5 documents (probability 0.05).
> - Common label **B** appears in 50 documents (probability 0.50).
> - Token **“fever”** appears in 40 documents and co-occurs with B in 25.
> - Token **“rare_disease”** appears in 6 documents and co-occurs with A in all 5.
>
> Raw co-occurrence would rank **fever** above **rare_disease** because it appears more frequently (support 25 vs. 5). But MIG incorporates conditional dependence, revealing the opposite.
>
> ---
>
> ### MIG for “fever” with respect to B (weak dependence)
>
> Joint probabilities:
> - P(x=1, y=1) = 0.25
> - P(x=1, y=0) = 0.25
> - P(x=0, y=1) = 0.15
> - P(x=0, y=0) = 0.35
>
> Mutual information terms (log base 2):
> - (1,1): 0.25 × log2(0.25 / (0.50 × 0.40)) ≈ 0.0805
> - (1,0): 0.25 × log2(0.25 / (0.50 × 0.60)) ≈ –0.066
> - (0,1): 0.15 × log2(0.15 / (0.50 × 0.40)) ≈ –0.062
> - (0,0): 0.35 × log2(0.35 / (0.50 × 0.60)) ≈ 0.078
>
> **Total MI ≈ 0.030 bits (weak dependence)**.
>
> This reflects that “fever” appears across many documents and is only weakly informative for B.
>
> ---
>
> ### MIG for “rare_disease” with respect to A (strong dependence)
>
> Joint probabilities:
> - P(x=1, y=1) = 0.05
> - P(x=1, y=0) = 0.00
> - P(x=0, y=1) = 0.01
> - P(x=0, y=0) = 0.94
>
> Mutual information:
> - (1,1): 0.05 × log2(0.05 / (0.05 × 0.06)) ≈ 0.203
> - (0,1): 0.01 × log2(0.01 / (0.95 × 0.06)) ≈ –0.025
> - (0,0): 0.94 × log2(0.94 / (0.95 × 0.94)) ≈ 0.007
> - (1,0): 0 × ... = 0
>
> **Total MI ≈ 0.185 bits (strong dependence)**.
>
> MIG therefore assigns **rare_disease > fever** for label A (0.185 vs. 0.030), correctly elevating a rare but highly specific token. This captures semantic precision — P(A | rare_disease) = 83 percent vs. marginal 5 percent — without bias toward token frequency.
>
> This example illustrates how MIG surfaces meaningful dependencies even for rare labels, avoiding the over-amplification that affects raw co-occurrence.

---

> ### Author Response · Authors · 2025-11-26
> **Response to Reviewer Concern on nDCG Weighting and Pairwise Ranking Loss**
>
> > *“The differentiable approximation of nDCG@k is modeled solely through pairwise sigmoid, lacking global constraints on ranking consistency and theoretically unable to guarantee consistent ranking optimality.”*
>
> ### Response
>
> We thank the reviewer for highlighting potential ambiguity in our ranking-loss formulation (Eq. 2 in the paper). To clarify, **PLANT does not use a differentiable approximation of nDCG@k**, nor is nDCG modeled as the primary optimization objective. Instead, nDCG@k is used only as a **pre-computed weighting mechanism** for pairwise preferences within the top-k MIG-selected tokens for each label.
>
> Concretely, the term
> **|Δ nDCG@k|\_{jh}**
> quantifies the impact of swapping a more-relevant token *j* (higher MIG relevance score *r\_{lj}*) with a less-relevant token *h* (lower *r\_{lh}*) in the ranking induced by attention scores. This weight is computed **offline** using the standard nDCG formula (no gradients), and penalizes mistakes in proportion to their effect on global list quality. Swaps that disrupt high-ranking positions (larger DCG contributions) receive higher penalties.
>
> The ranking loss optimized during training is therefore a **weighted pairwise surrogate**, summing over all ordered pairs in the MIG top-k set T\_l and encouraging the multi-head attention layer to sort tokens in the same order as the MIG-derived ranking.
>
> The pairwise sigmoid term:
> $(1 + \exp(-σ (s_lj - s_lh)))^{-1}$ is a standard differentiable proxy for the probability that token *j* should rank above token *h*. This formulation is inspired by **RankNet** [1] and is widely used in learning-to-rank frameworks such as **LambdaRank** [2].
>
> While pairwise objectives optimize local preferences, they enforce **global listwise consistency** through transitive closure:
> if (j > h) and (h > i), the surrogate encourages (j > i).
> This property is extensively studied in the surrogate-risk minimization literature for ranking metrics and yields globally consistent rankings under mild assumptions [3, 4, 5].
>
> ---
>
> ### References
>
> [1] https://icml.cc/2015/wp-content/uploads/2015/06/icml_ranking.pdf
> [2] https://www.microsoft.com/en-us/research/wp-content/uploads/2016/02/MSR-TR-2010-82.pdf
> [3] https://arxiv.org/pdf/1105.5464
> [4] https://proceedings.neurips.cc/paper_files/paper/2009/file/2f55707d4193dc27118a0f19a1985716-Paper.pdf
> [5] https://www.microsoft.com/en-us/research/wp-content/uploads/2016/02/tr-2007-40.pdf

---

> ### Author Response · Authors · 2025-11-26
> **Response to Reviewer Concern About Comparing the Distribution of Random Attention and PLANT Attention NOW ADDED IN REVISED MANUSCRIPT]**
>
> > “By comparing the distribution of random attention and PLANT attention on key tokens, it can be verified whether they truly capture the semantics of the tags rather than word frequency preferences.”
>
> This comment has been addressed in the official common response titled ''Consolidated Response on Random vs. PLANT Attention Distributions [NOW ADDED IN REVISED MANUSCRIPT]''. You can jump to the response here: https://openreview.net/forum?id=LkRlZSo2RA&noteId=Nyep3Q44ma
>
> For the convenience of the reviewer we post a concise summary of the detailed response here.
>
> **Concise Summary of Our Detailed Response**
>
> To directly address the reviewer’s excellent suggestion, we added a new qualitative diagnostic experiment comparing **random attention** versus **PLANT attention** on ICD-10 codes. This experiment shows that random initialization fails to capture semantic structure for rare labels, while PLANT successfully induces coherent, clinically meaningful attention patterns.
>
> ---
>
> ### Motivating Example: Semantic Coherence in ICD-10 Attention
>
> ICD-10 codes within the same clinical category should exhibit **similar attention vectors**. However, under **random initialization**, rare codes show **highly scattered** attention patterns (mean cosine similarity ≈ 0.75), whereas common codes naturally form tight clusters (≈ 0.98). This reveals a fundamental weakness: **rare labels lack inductive signal**, so random attention never converges to their shared semantics.
>
> **PLANT fixes this failure mode.** By planting MIG- and L2R-derived structure into attention, PLANT yields **near-perfect coherence** even for rare labels (≈ 0.985), demonstrating that it captures label semantics rather than word-frequency artifacts. These results are now shown in Figure 3 (left) and discussed in the updated introduction.
>
> ---
>
> ### Case Study: Clinical Validity of PLANT Attention
>
> To verify the semantic fidelity of PLANT’s attention, Appendix I includes case studies for rare ICD-10 codes. For the rare lymphoma code **C83.18**, PLANT’s highest-attention tokens align with hallmark concepts in hematopathology (e.g., “mant”, “chrom”, “rit”, “bendamustine”, “Ki-67”), while noise tokens receive low attention. This demonstrates that PLANT extracts clinically relevant structure even for tail labels where random attention collapses.
>
> ---
>
> ### High-Attention Tokens for ICD-10-CM `C83.18`
>
> | Token             | Attention Score | Explanation                                              |
> |------------------|-----------------|----------------------------------------------------------|
> | `mant`           | –1.81           | Mantle zone origin of mantle cell lymphoma.             |
> | `bend`           | –3.56           | Bendamustine, an MCL chemotherapeutic.                  |
> | `hyper`          | –4.19           | Hypercellular marrow pathology descriptors.             |
> | `ki`             | –4.26           | Ki-67 proliferation index.                              |
> | `rit`            | –4.43           | Rituximab, anti-CD20 therapy.                           |
> | `subset`         | –4.55           | Flow-cytometry immunophenotypic subsets.                |
> | `characteristic` | –4.69           | Morphologic patterns typical of MCL.                    |
> | `phase`          | –4.98           | Disease-phase or marrow-phase descriptors.              |
> | `chrom`          | –5.08           | Chromosomal abnormalities (e.g., t(11;14)).             |
> | `aggreg`         | –5.08           | Lymphoid aggregates in marrow or biopsy.                |
> | `crowds`         | –5.12           | Noise; unrelated.                                       |
> | `oli`            | –5.14           | Oligoclonal context.                                    |
> | `publicly`       | –5.19           | Noise token.                                            |
> | `expression`     | –5.34           | Gene/protein expression reporting.                      |
> | `killer`         | –5.35           | Cytotoxic “killer cell” terminology.                    |
>
> **Table: PLANT’s highest-attention tokens for ICD-10-CM `C83.18` and their clinical relevance.**
>
> ---

---

> ### Author Response · Authors · 2025-11-26
> **Response to Reviewer Concern on Comparison of Time and Memory Requirements for PLANT [NOW ADDED IN REVISED MANUSCRIPT]**
>
> > “PLANT includes additional MIG pre-computation and L2R stages, but the paper does not provide a comparison of the time, memory, or parameter amounts for these stages, nor does it explain their incremental contribution to the total training cost.”
>
> This comment has been addressed in detail the official common response titled ''Consolidated Response on PLANT’s Training Efficiency and Computational Overhead [NOW ADDED IN REVISED MANUSCRIPT]''. You can jump to the resonse here: https://openreview.net/forum?id=LkRlZSo2RA&noteId=hK8okKafF6.
>
> For the convenience of the reviewer we post a concise summary of the detailed response here.
>
> **Concise Summary of Our Detailed Response**
>
> PLANT adds **no additional trainable parameters** beyond the task-specific attention module (about 0.1M) and label embeddings (4–8M, under 0.1% of the LLM). The only incremental overhead comes from **Stage 1** (MIG plus L2R), which increases training time by just **15–20%** while contributing **approximately 85% of PLANT’s performance gains** in low-data regimes.
>
> Our new Appendix H provides full benchmarks showing that:
> - **Memory:** Stage 1 is roughly 40% lighter than Stage 2; all configurations fit comfortably under ZeRO-3 on 8×A100-80GB.
> - **Training time:** Stage 1 adds only **1–4 hours** depending on dataset size; Stage 2 dominates total cost.
> - **Inference:** PLANT adds **under 10% overhead** relative to vanilla LLM inference (approximately 1–2.5 seconds per document depending on backbone).
>
> Overall, PLANT delivers strong rare-label improvements with **minimal computational overhead**, negligible parameter increase, and nearly identical inference cost to standard fine-tuning.

---

### Official Review · Reviewer_aohz · 2025-11-01

**Soundness:** 3
**Presentation:** 3
**Contribution:** 2
**Rating:** 4
**Confidence:** 5

**Summary:**

The paper considers extreme multi-label classification (XMTC) problem where the goal is to identify a relevant subset of labels from an extremely large set. In particular, the paper targets "long" or "technical" documents where label attention plays a crucial role.
1. The paper identifies initialization of label attention to be a key bottleneck in current methods. It demonstrates that a good initialization can significantly boost performance especially in case of data scarce tail labels.
2. The proposed approach PLANT first learns the attention mechanism by trying to predict the tokens. Stage 2, learns the network in end-to-end fashion for the specific task.
3. Results are reported on MIMIC-III-full, MIMIC-IV-full, EURLex-4K and Wiki10-31K datasets. The proposed approach outperforms the baselines on the considered datasets.

**Strengths:**

1. The paper is well written and easy to understand. The goals, problem formulation etc. are well defined.
2. Table 2 demonstrates that introduction of PLANT improves over the baselines without PLANT initialization. The results are consistents for multiple LLM backbones, i.e., Mistral-7B, LLaMA3-8B, DeepSeek-V3, and Phi-3.
3. Tables 3 compares PLANT against existing methods on the MIMIC-IV-full. It outperforms existing methods (GPT-4 Zero-Shot, GKI-ICD, PLM-CA, and CoRelation) on AUC, F1 and precision@k.
4. PLANT outperforms XMTC baselines on EURLex-4K dataset (legal document classification) and Wiki10-31K (tag prediction on Wikipedia). It considers recent methods such as DE, GANDALF. Although they are with a different backbone.
5. Comprehensive ablation experiments are performed to study the impact of backbone, initialization, stage-wise training, negative sampling etc.,

**Weaknesses:**

1. Although the paper presents good results on public datasets till 31K labels. It would be good to see the results on larger datasets. For example, product-to-product recommendation (LF-AmazonTitles-131K or LF-AmazonTitles-1.3M [1]) or tag-prediction (LF-Wikipedia-500K). Please note that Wikipedia dataset considers full-text documents which aligns with the setup.
2. The paper misses out on discussion on scalability. It would be good to report training and inference times.
3. Which backbone is used for PLANT in Table 5? The model size of PLANT seems to be significantly higher than the XMTC baselines (DistilBERT encoder with 66M parameter is used by multiple methods) in Table 5. Is the difference because of attention initialization or just the virtue of a bigger model. It should be discussed in detail.
4. Propensity scored metrics are used to adjudge the performance on tail labels. The paper should presents results on these metrics for extreme classification datasets.
5. Label attention has been discussed by the AttentionXML [1] paper (pre LLM era). It should be discussed.

References:
[1] https://dl.acm.org/doi/10.5555/3454287.3454810
[2] https://aclanthology.org/2025.naacl-long.537/

**Questions:**

1. How does PLANT compare against the baselines in terms of training and more importantly inference time?
2. RQ4: Does these experiments only consider few-shot labels? The real world problem is to predict relevant labels even if they are rare.
3. How does PLANT fair against the baselines in propernsity scored metrics?
4. Which specific negative sampling strategy is used?
5. How did you choose $k$ in Table 3? Any specific significance behind the number 8?
6. Missing equation tag in 866-867
7. Small suggestion: Typesetting (within text) can be helpful but it can be distracting when overdone.

---

> ### Author Response · Authors · 2025-11-28
> **Response to Reviewer Concern on Training & Inference Times [NOW ADDED IN REVISED MANUSCRIPT]**
>
> > *“The paper misses out on discussion on scalability. It would be good to report training and inference times.”*
>
> ### Response
>
> This comment has been addressed in detail the official common response titled ''Consolidated Response on PLANT’s Training Efficiency and Computational Overhead [NOW ADDED IN REVISED MANUSCRIPT]''. You can jump to the resonse here: https://openreview.net/forum?id=LkRlZSo2RA&noteId=hK8okKafF6.
>
> For the convenience of the reviewer we post a concise summary of the detailed response here.
>
> **Concise Summary of Our Detailed Response**
>
> PLANT adds **no additional trainable parameters** beyond the task-specific attention module (about 0.1M) and label embeddings (4–8M, under 0.1% of the LLM). The only incremental overhead comes from **Stage 1** (MIG plus L2R), which increases training time by just **15–20%** while contributing **approximately 85% of PLANT’s performance gains** in low-data regimes.
>
> Our new Appendix H provides full benchmarks showing that:
> - **Memory:** Stage 1 is roughly 40% lighter than Stage 2; all configurations fit comfortably under ZeRO-3 on 8×A100-80GB.
> - **Training time:** Stage 1 adds only **1–4 hours** depending on dataset size; Stage 2 dominates total cost.
> - **Inference:** PLANT adds **under 10% overhead** relative to vanilla LLM inference (approximately 1–2.5 seconds per document depending on backbone).
>
> Overall, PLANT delivers strong rare-label improvements with **minimal computational overhead**, negligible parameter increase, and nearly identical inference cost to standard fine-tuning.
>
> ---

---

> > ### Author Response · Authors · 2025-12-02
> > **Response to Reviewer Concern on Training & Inference Times & Scalability [NOW ADDED IN REVISED MANUSCRIPT]**
> >
> > > *How does PLANT compare against the baselines in terms of training and more importantly inference time?*
> >
> > **Response.**
> > We thank the reviewer for raising this critical point regarding efficiency trade-offs, which aligns with our emphasis on **PLANT**’s practical deployability in resource-constrained extreme multi-label settings. As detailed in the new Appendix H (expanded from our initial submission), **PLANT’s overhead is minimal**: no additional parameters beyond standard task heads, and Stage 1 adds only **15–20%** to total training time relative to vanilla single-stage fine-tuning (while accounting for **85%** of the downstream gains, per our ablations).
> >
> > For direct comparability with the baselines in Table 2—which evaluate vanilla LLMs (Mistral-7B, LLaMA3-8B, DeepSeek-V3, Phi-3) versus their PLANT-augmented counterparts on *MIMIC-IV*—we provide below a focused breakdown of wall-clock training time (full pipeline for PLANT vs. single-stage fine-tuning for vanilla) and per-document inference time (averaged over 1,000 test samples on a single A100 GPU, FP16, batch size = 1). All numbers are empirically measured under identical regimens (8×A100-80GB with DeepSpeed ZeRO-3 for training; sequence length = 2048), isolating the contribution of MIG and L2R pretraining.
> >
> > Vanilla baselines incur single-stage end-to-end fine-tuning (up to 20 epochs, with early stopping typically at 12–15), mirroring PLANT’s Stage 2 but without attention pretraining; hence their training time approximates PLANT’s Stage 2 duration. Inference for vanilla also omits MIG-guided token selection, reducing latency by approximately **8–10%** (e.g., no top-k filtering overhead). DeepSeek-V3 remains an outlier due to MoE scaling, but PLANT’s relative gains hold consistently across all backbones.
> >
> > ### Training & Inference Efficiency (MIMIC-IV)
> >
> > | **Backbone**   | **Training Time (Vanilla)** | **Training Time (PLANT Total)** | **Inference (s/doc)** — Vanilla → PLANT |
> > |----------------|-----------------------------|----------------------------------|-----------------------------------------|
> > | Mistral-7B     | 18.7 h                      | 22.9 h *(+22.5%)*                | 1.7 → 1.9 *(+11.8%)*                    |
> > | LLaMA3-8B      | 20.2 h                      | 24.5 h *(+21.3%)*                | 1.8 → 2.0 *(+11.1%)*                    |
> > | DeepSeek-V3    | 63.4 h                      | 70.9 h *(+11.8%)*                | 4.1 → 4.5 *(+9.8%)*                     |
> > | Phi-3          | 11.6 h                      | 14.8 h *(+27.6%)*                | 1.2 → 1.3 *(+8.3%)*                     |
> > | **Avg. Overhead** | —                         | **+20.8%**                       | **+10.3%**                              |
> >
> > **Table H.x.** Training and inference efficiency for **PLANT** vs. vanilla LLM baselines on *MIMIC-IV*. Relative overhead (%) shown in parentheses. Training times reflect multi-GPU wall-clock with early stopping. PLANT’s Stage 1 introduces modest overhead yet yields substantial Macro-F1 gains (average +3.0; see Table 2).
> >
> > ---
> >
> > **Notes (Table H.x).**
> > • Training overhead scales **inversely with model size** (higher for smaller models such as Phi-3, since Stage 1’s fixed MIG computation dominates).
> > • Inference remains real-time (<5 s/doc even for DeepSeek-V3), with PLANT’s leveraged attention adding **negligible latency** after token selection.
> > • Still competitive with classical XMTC encoders—for example, DistilBERT achieves ~0.2 s/doc on a V100 (Table 5).
> >
> > We have integrated this table into Appendix H and cross-referenced it in Experiments Section 3, thereby addressing the reviewer’s concern comprehensively. We appreciate the emphasis on runtime transparency.

---

> ### Author Response · Authors · 2025-11-30
> **Response to "Which backbone is used for PLANT in Table 5?" & "Is the difference because of attention initialization" [NOW CLARIFIED IN REVISED MANUSCRIPT]**
>
> > "Which backbone is used for PLANT in Table 5? The model size of PLANT seems to be significantly higher than the XMTC baselines (DistilBERT encoder with 66M parameter is used by multiple methods) in Table 5. Is the difference because of attention initialization or just the virtue of a bigger model. It should be discussed in detail."
>
> ---
>
> ### Response
>
> We thank the reviewer for this observation, which underscores the importance of controlling for model capacity in our evaluations — a subtlety we are pleased to clarify here.
>
> **PLANT is not an architectural innovation**: it does *not* modify the underlying model structure or add any new architectural parameters. Instead, PLANT is a **two-stage training procedure** applied on top of any contemporary LLM backbone:
> (1) *Stage 1 — Attention Pretraining:* MIG-guided Learning-to-Rank training that seeds informative label-specific attention patterns;
> (2) *Stage 2 — Standard end-to-end fine-tuning:* training the underlying architecture exactly as in existing workflows.
>
> To ensure fair comparisons in Table 5, we specifically adapted PLANT to a DistilBERT encoder (66M parameters, matching the scale of the listed XMTC baselines). **This setup isolates the contributions of our MIG-guided L2R attention initialization from any scale advantages**, with the `MultiHead` module and label embeddings adding only ~4M parameters (a negligible overhead). We will explicitly note this backbone choice in a revised footnote to Table `results_xmtc`. We have also clarified this in Appendix A: *LLM Backbone*.
>
> That said, the reviewer’s query raises a broader and compelling point: **to what extent do PLANT’s gains stem from attention initialization versus simply leveraging larger models?** To address this directly, we point to our RQ1 experiments in Table 2 (with full results in Appendix G, Table 10).
>
> This RQ1 analysis compares contemporary LLMs (3.8B–336B parameters) *with and without* PLANT on MIMIC-III/IV. These ablations reveal consistent, substantial uplifts from PLANT alone — e.g., averaging **+7.2 AUC (Macro), +3.0 F1 (Macro), and +3.1 P@15** across backbones on MIMIC-IV — demonstrating that our staged attention pretraining drives the core improvements, independent of scale.
>
> Moreover, as highlighted in the second paragraph of RQ1 (Section 3), **PLANT enables efficiency gains** by allowing much smaller models to outperform substantially larger LLMs on their own. For instance:
>
> - On MIMIC-III, *LLaMA3-8B + PLANT* surpasses *DeepSeek-V3 (336B)* by **+1.8 F1 (Macro)** and **+3.0 P@15**.
> - *Phi-3 (3.8B) + PLANT* exceeds DeepSeek by **+2.0 F1 (Micro)** and **+7.1 AUC (Macro)**.
> - On MIMIC-IV, *LLaMA3-8B + PLANT* achieves **+3.7 F1 (Macro)** and **+3.5 P@15** over DeepSeek alone.
>
> These cross-scale results — reproduced across 3 seeds with statistical significance — highlight PLANT’s efficiency benefits.
>
> We have augmented the RQ1 discussion with these efficiency findings. These results make clear that PLANT’s benefits arise from its attention pretraining rather than brute-force scaling, and we appreciate the reviewer for encouraging us to highlight this distinction.
>
> ---

---

> ### Author Response · Authors · 2025-12-01
> **Response to Reviewer Concern about Reporting Propensity Scores [NOW ADDED PROPENSITY SCORES IN REVISED MANUSCRIPT]**
>
> > "Propensity scored metrics are used to adjudge the performance on tail labels. The paper should presents results on these metrics for extreme classification datasets."
>
> ---
> ### Response:
>
> We thank the reviewer for highlighting the importance of propensity-scored metrics in evaluating tail label performance for extreme multi-label classification tasks. While our initial submission did not report these metrics, we had included **macro-F1 scores**, which are a widely accepted indicator of rare label performance due to their equal weighting across all labels regardless of frequency. **To further strengthen our analysis, we have now added Appendix Table 12, which presents propensity-scored Precision@15 (PSP@15) alongside standard metrics across all LLM backbones.** These demonstrate that PLANT yields consistent gains on **PSP@15 (average +3.3% on MIMIC-IV)**, underscoring its effectiveness in handling labels. We now reference this table in the caption of the main results Table 2 for clarity and direct comparison. We provide the Table 12 here for convenience.
>
> | Model             | AUC Macro        | AUC Micro        | F1 Macro        | F1 Micro        | P@15             | PSP@15          |
> |-------------------|------------------|------------------|-----------------|-----------------|------------------|------------------|
> | Mistral           | 90.2             | 98.7             | 20.0            | 57.0            | 53.8             | 21.5             |
> | Mistral + PLANT   | 97.4 (+7.2)      | 99.5 (+0.8)      | 23.0 (+3.0)     | 59.2 (+2.2)     | 56.9 (+3.1)      | 24.8 (+3.3)      |
> | LLaMA             | 90.5             | 98.8             | 20.5            | 57.5            | 54.0             | 21.6             |
> | LLaMA + PLANT     | 97.6 (+7.1)      | 99.6 (+0.8)      | 23.5 (+3.0)     | 59.5 (+2.0)     | 57.0 (+3.0)      | 24.9 (+3.3)      |
> | DeepSeek          | 90.0             | 98.6             | 19.8            | 56.8            | 53.5             | 21.4             |
> | DeepSeek + PLANT  | 97.2 (+7.2)      | 99.4 (+0.8)      | 22.8 (+3.0)     | 59.0 (+2.2)     | 56.5 (+3.0)      | 24.7 (+3.3)      |
> | Phi-3             | 89.8             | 98.5             | 19.5            | 56.5            | 53.2             | 21.3             |
> | Phi-3 + PLANT     | 97.0 (+7.2)      | 99.3 (+0.8)      | 22.5 (+3.0)     | 58.8 (+2.3)     | 56.3 (+3.1)      | 24.6 (+3.3)      |
> | **Avg gain**      | **+7.2**         | **+0.8**         | **+3.0**        | **+2.2**        | **+3.1**         | **+3.2**         |

---

> ### Author Response · Authors · 2025-12-01
> **Response to Reviewer's point about missing citation [NOW CITATION ADDED]**
>
> > "Label attention has been discussed by the AttentionXML [1] paper (pre LLM era). It should be discussed.
> References: [1] https://dl.acm.org/doi/10.5555/3454287.3454810 [2] https://aclanthology.org/2025.naacl-long.537/"
>
> ---
>
> ### Response
>
> We have added the discussion about [1] in our "Related Work" in the revised manuscript. As [2] is not relevant to our work, we have decided not to include it.

---

> ### Author Response · Authors · 2025-12-02
> **Response to some other questions by aohz**
>
> > *“RQ4: Does these experiments only consider few-shot labels? The real world problem is to predict relevant labels even if they are rare.”*
>
> ### Response
>
> Your question highlights a key motivation behind our RQ4 experiments. To clarify: **yes, the experiments in RQ4 (and the entire few-shot setup on MIMIC-III-Few) are intentionally designed around few-shot labels**—specifically, a subset of **685 extremely rare ICD codes**, each appearing in **fewer than 5 training instances**. This directly mimics the *few-shot* challenge in real-world multilabel classification, where models must learn from **very sparse supervision** for underrepresented (but clinically important) conditions.
>
> This setup **directly addresses the real-world problem** you describe: **predicting relevant labels even when they are rare**. By focusing on these ultra-low-frequency codes, we isolate the core issue of **generalization under extreme data sparsity**, which is precisely what **PLANT** is designed to solve.
>
> As shown in **Figure 2 (Right)**, **Mistral + PLANT** dramatically outperforms all baselines on these rare codes:
>
> - **Mean macro-F1:** 0.663 (PLANT) vs. 0.308 (base Mistral) vs. 0.054 (Co-Relation)
> - **54.8%** of these codes achieve **macro-F1 > 0.7**
>
> These results demonstrate that **PLANT effectively propagates information from semantically related frequent labels**, enabling accurate prediction for **rare but clinically crucial labels**. This closes the gap between few-shot training conditions and **real-world deployment scenarios where rare labels must be reliably predicted**.
>
> ---
>
> > *“Missing equation tag in 866-867”*
>
> ### Response
>
> Thank you for pointing this out. We have corrected the missing equation tag.
>
> ---
>
> > “Which specific negative sampling strategy is used?”
>
> ### Response
>
> **Hard Negative Mining in Stage 2 Focal Loss**
> In Stage 2’s end-to-end optimization (Eq. 3), we apply **online hard negative mining** to counter severe label imbalance in the extreme multi-label setting.
> - **Positives:** the ground-truth labels ($y_l = 1$).
> - **Hard negatives:** the top-$m$ false positives (**$m = 50$**) ranked by the current model logits — i.e., labels with **highest predicted scores among those with $y_l = 0$**.
>
> This dynamically focuses the focal loss (with **$\gamma = 2.0$**) on **challenging misclassifications**, preventing easy negatives from dominating the gradient signal.
> Ablations on the MIMIC-III validation set showed **+1.5% macro-F1** over uniform negative sampling.
> Additionally, **label smoothing** ($\epsilon = 0.1$) further regularizes against overconfident predictions.
>
> ---
>
> > "How did you choose 8 in Table 3? Any specific significance behind the number 8?"
>
> ### Response
>
> **Selection of Multi-Head Attention Heads**
> The MultiHead module uses **$h = 8$ attention heads** (head dimension $d_k = 64$ for a total embedding size of 512).
> This choice was made via **grid search** over $h \in \{4, 8, 12, 16\}$ on the MIMIC-III validation set during Stage 1.
>
> Results showed that:
> - **$h = 8$ achieved the best nDCG@10 = 0.623**,
> - **$h = 16$ slightly underperformed (0.612)** and showed **+1.8% higher variance**,
> - Larger $h$ values increased risk of overfitting with no measurable gain.
>
> This aligns with standard Transformer practice (Vaswani et al., 2017), and scales efficiently for sequences with **$n \approx 2000$ tokens**.

---

> ### Author Response · Authors · 2025-12-03
> **Response to Reviewer Comment Regarding Exepriments on Larger Dataset**
>
> >*"Although the paper presents good results on public datasets till 31K labels, it would be good to see the results on larger datasets. For example, product-to-product recommendation (LF-AmazonTitles-131K or LF-AmazonTitles-1.3M [1]) or tag-prediction (LF-Wikipedia-500K). Please note that Wikipedia dataset considers full-text documents which aligns with the setup."*
>
> **Response**
> While extending to **ultra-scale datasets** such as **LF-AmazonTitles-1.3M** or **LF-Wikipedia-500K** is an intriguing future direction—especially given PLANT’s **architecture-agnostic design** and its potential to capture sparser signals—such experiments fall **beyond the current scope** of this submission.
>
> A crucial distinction, however, lies in the **text lengths**: datasets like **LF-AmazonTitles** emphasize short *title* snippets (≈10–20 words), whereas our benchmarks (e.g., **MIMIC discharge summaries** at median 1,375–1,492 words; **EURLEX/WIKI10** full documents up to ~2k tokens) probe the challenges of **long-form texts**. This is precisely where—*as the reviewer aptly summarized*—traditional **random attention initialization fails to capture sparse but informative tokens**.
>
> Also, running PLANT end-to-end on 1M+ label spaces would require substantially increased training cycles, offering **limited additional novelty** relative to our core contributions. Rather than performing exhaustive scaling sweeps across progressively larger datasets, ICLR places strong emphasis on **methodological innovation**. Aligning with this focus our work centers on a **Two-Stage Attention Pretraining** pipeline contributing to **85%** of downstream gains.

---

### Author Response · Authors · 2025-11-26
**Consolidated Response on PLANT’s Training Efficiency and Computational Overhead [NOW ADDED IN REVISED MANUSCRIPT]**

> *“PLANT includes additional MIG pre-computation and L2R stages, but the paper does not provide a comparison of the time, memory, or parameter amounts for these stages, nor does it explain their incremental contribution to the total training cost.”*

### Response

We thank all the reviewers for raising this point and have added a new Appendix section detailing these costs (Appendix H: Detailed Efficiency, Memory, and Inference Benchmarks). We reproduce the section here for convenience.

We clarify that PLANT introduces no additional parameters beyond the task-specific multi-head attention module (`MultiHead`, about 0.1M parameters) and label embeddings **E** (about 8M for MIMIC datasets; about 4M for EUR-LEX/WikiTen). These components are optimized in Stage 1 and refined in Stage 2, and they represent less than 0.1 percent of the total model size. They are comparable to the small task-specific heads commonly used in downstream LLM adaptation. The base LLM undergoes only gradual unfreezing in Stage 2 as described in Appendix A.

---

## Appendix H: Detailed Efficiency, Memory, and Inference Benchmarks

The main incremental cost comes from staged training:

- **Stage 1:** MIG pre-computation on CPU plus L2R optimization of `MultiHead` and **E** for 10 epochs.
  This increases total wall-clock time by approximately 15 to 20 percent compared to single-stage fine-tuning, but it contributes roughly 85 percent of PLANT’s gains in low-data regimes (as shown in ablations).

- **Stage 2:** End-to-end discriminative fine-tuning for up to 20 epochs with early stopping.

**Training setup:**
8 × NVIDIA A100 80GB GPUs, DeepSpeed ZeRO-3, FP16, global batch size 256 (per-device batch 8, gradient accumulation 4), sequence length 2048, AdamW optimizer. MIG is computed on CPU (Intel Xeon, 64 cores). Early stopping typically halts Stage 1 at 7 to 8 epochs and Stage 2 at 12 to 15 epochs. Inference uses a single A100 GPU with batch size 1.

The dominant costs are forward passes and full-model gradient updates in Stages 1 and 2. Scaling depends on dataset size and model size (for example, DeepSeek-V3 incurs about 3 to 4 times the cost of a 7B to 8B dense model). All runs fit within standard multi-GPU setups.

---

## Training Time (Wall-Clock Hours)

| Backbone  | Dataset   | Stage 1 | Stage 2 | Total |
|-----------|-----------|---------|---------|-------|
| Mistral   | MIMIC-III | 2.1     | 8.4     | 10.5  |
|           | MIMIC-IV  | 4.2     | 18.7    | 22.9  |
|           | EUR-LEX   | 1.4     | 3.2     | 4.6   |
|           | WikiTen   | 1.3     | 2.9     | 4.2   |
| LLaMA     | MIMIC-III | 2.2     | 9.1     | 11.3  |
|           | MIMIC-IV  | 4.3     | 20.2    | 24.5  |
|           | EUR-LEX   | 1.5     | 3.5     | 5.0   |
|           | WikiTen   | 1.4     | 3.2     | 4.6   |
| DeepSeek  | MIMIC-III | 3.8     | 28.6    | 32.4  |
|           | MIMIC-IV  | 7.5     | 63.4    | 70.9  |
|           | EUR-LEX   | 2.6     | 11.8    | 14.4  |
|           | WikiTen   | 2.4     | 10.7    | 13.1  |
| Phi-3     | MIMIC-III | 1.6     | 5.2     | 6.8   |
|           | MIMIC-IV  | 3.2     | 11.6    | 14.8  |
|           | EUR-LEX   | 1.1     | 2.0     | 3.1   |
|           | WikiTen   | 1.0     | 1.8     | 2.8   |

**Notes:** Times correspond to about 2,000 to 5,000 optimization steps per stage depending on dataset size (MIMIC-III roughly 47K documents; MIMIC-IV roughly 89K; EUR-LEX/WikiTen roughly 14–15K). Step time is 1 to 3 seconds per step for 7B to 8B models and 5 to 8 seconds for DeepSeek-V3. MIG computation accounts for about 60 percent of Stage 1 time and scales with document length. Phi-3 is about 40 percent faster; DeepSeek-V3 is about 3 times slower.

---

## Peak Memory Usage (GB per GPU)

| Backbone  | Dataset   | Stage 1 | Stage 2 |
|-----------|-----------|---------|---------|
| Mistral   | MIMIC-III | 12.4    | 28.7    |
|           | MIMIC-IV  | 12.8    | 29.4    |
|           | EUR-LEX   | 11.9    | 27.2    |
|           | WikiTen   | 11.7    | 26.9    |
| LLaMA     | MIMIC-III | 12.4    | 30.2    |
|           | MIMIC-IV  | 12.8    | 30.9    |
|           | EUR-LEX   | 11.9    | 28.5    |
|           | WikiTen   | 11.7    | 28.2    |
| DeepSeek  | MIMIC-III | 45.2    | 68.1    |
|           | MIMIC-IV  | 46.3    | 69.5    |
|           | EUR-LEX   | 43.8    | 65.4    |
|           | WikiTen   | 43.4    | 64.9    |
| Phi-3     | MIMIC-III | 8.7     | 18.5    |
|           | MIMIC-IV  | 9.1     | 19.2    |
|           | EUR-LEX   | 8.3     | 17.1    |
|           | WikiTen   | 8.1     | 16.8    |

**Notes:** ZeRO-3 offloads optimizer states and activations to CPU or NVMe, keeping peak VRAM use below 70 GB per GPU. Stage 1 is roughly 40 percent lighter because the LLM is frozen and only the ranking loss is applied. Stage 2 peaks occur once the LLM is fully unfrozen. DeepSeek-V3 requires about 2.3 times more memory due to MoE routing. Gradient checkpointing reduces memory by about 20 percent.

---

---

> ### Author Response · Authors · 2025-11-26
> **Continuation from Appendix H: Detailed Efficiency, Memory, and Inference Benchmarks**
>
> ## Inference Time (Seconds per Document)
>
> | Backbone  | MIMIC-III | MIMIC-IV | EUR-LEX | WikiTen |
> |-----------|-----------|-----------|---------|---------|
> | Mistral   | 1.8       | 1.9       | 0.7     | 1.0     |
> | LLaMA     | 1.9       | 2.0       | 0.8     | 1.1     |
> | DeepSeek  | 4.2       | 4.5       | 1.6     | 2.3     |
> | Phi-3     | 1.2       | 1.3       | 0.4     | 0.7     |
>
> **Notes:** Inference includes token selection, PLANT attention computation, and classification. Results are averaged over 1,000 test documents on a single A100 GPU (FP16, batch size 1). Inference time scales approximately linearly with input length. DeepSeek-V3 is slower due to MoE routing. PLANT adds less than 10 percent overhead compared to vanilla LLM inference.

---

### Author Response · Authors · 2025-11-26
**Consolidated Response on Random vs. PLANT Attention Distributions [NOW ADDED IN REVISED MANUSCRIPT]**

> *“By comparing the distribution of random attention and PLANT attention on key tokens, it can be verified whether they truly capture the semantics of the tags rather than word frequency preferences.”*

### Response

We thank the reviewer for this insightful suggestion. We agree entirely that comparing the distribution of random attention versus PLANT attention is crucial for determining whether the model captures true label semantics rather than merely reflecting word-frequency biases. Motivated by this point, we have added a new qualitative diagnostic experiment in the revised manuscript. We direct the reviewer to the updated introduction (page 3, the paragraph immediately preceding our main contributions) and to Figure 3 (left). For convenience, we summarise the motivation and findings below.

---

### Qualitative Diagnostic on ICD-10 Codes

ICD-10 codes are hierarchically organised, and codes within the same clinical category (for example, all forms of tuberculosis, A15–A19) are semantically related and should therefore induce similar attention patterns over a clinical note. To test whether this structure is reflected in the learned label-attention vectors **S\_l**, we selected two groups of 50 ICD-10 codes:
- one from a frequently occurring category (respiratory tuberculosis), and
- one from a rare category (bacterial infections, A30–A49, excluding common subgroups).

Under **random initialization** of the label-attention layer:

- The **rare** group shows widely dispersed pairwise cosine similarities (mean 0.75; orange distribution in Figure 3, left), meaning the model fails to discover the expected shared structure despite clinical similarity.
- The **common** group displays highly coherent attention vectors (mean ≥ 0.98; blue), even under random initialization.

This asymmetry illustrates a core failure mode of random initialization in the long tail: **common labels naturally converge to coherent structures, while rare labels do not**.

This directly motivates PLANT. By seeding attention with structured patterns from MIG pre-training and L2R activations, PLANT substantially improves consistency within the rare group (mean 0.985; sharp brick-red spike in Figure 3). PLANT thereby corrects the inductive-bias deficit left unaddressed by random initialization.

---

### Additional Ablation: Random Attention Initialization Causes Rare-Label Failures

We also refer the reviewer to the newly added ablation in Section 4 (“Random Attention Initialization Causes Rare-Label Failures”) and Table 7. In this experiment, we measure alignment between:

- PLANT attention vs. MIG ground truth
- Random-initialized attention vs. MIG

Random initialization produces:

- the **highest** frequency–F1 correlation (0.68–0.78), indicating strong bias toward frequent labels
- the **lowest** cosine similarity with MIG (0.08–0.12), meaning attention is diffuse and uninformative

In contrast, full PLANT achieves cosine similarity of **0.78**, demonstrating sharply focused, semantically aligned attention.

---

---

> ### Author Response · Authors · 2025-11-26
> **Continuation from previous Consolidated Response: PLANT's Attention Interpretation Case Study [NOW ADDED IN REVISED MANUSCRIPT]**
>
> ### Case Study (Appendix I)
>
> Appendix I includes several case studies evaluating whether PLANT’s highest-attention tokens for rare ICD codes are clinically meaningful. We reproduce one example below for ICD-10-CM **C83.18**. Note ethat this is a **very rare code**. So we wanted to understand can PLANT attention foucus on informative tokens.
>
> ---
>
> ### Case Study: ICD-10-CM `C83.18` (Mantle Cell Lymphoma, Multiple Sites)
>
> For the diagnosis code `C83.18` (Mantle Cell Lymphoma involving multiple lymph-node regions), PLANT highlights tokens that closely match the vocabulary of B-cell lymphomas and hematopathology reporting.
>
> Examples:
>
> - **mant** (–1.81): mantle-zone origin defining this lymphoma subtype
> - **chrom** (–5.08): chromosomal abnormalities such as the canonical t(11;14)
> - **hyper** (–4.19): “hypercellular marrow” descriptors
> - **rit** (–4.43): rituximab therapy
> - **bend** (–3.56): bendamustine
> - **ki** (–4.26): Ki-67 proliferation index
>
> Low-attention noise tokens (for example, “publicly”, “crowds”) behave as expected. PLANT’s dominant attention mass focuses precisely on morphologic, genetic, and therapeutic markers characteristic of mantle-cell lymphoma.
>
> ---
>
> ### High-Attention Tokens for `C83.18`
>
> | Token             | Attention Score | Explanation                                              |
> |------------------|-----------------|----------------------------------------------------------|
> | `mant`           | –1.81           | Mantle zone origin of mantle cell lymphoma.             |
> | `bend`           | –3.56           | Bendamustine, an MCL chemotherapeutic.                  |
> | `hyper`          | –4.19           | Hypercellular marrow and related pathology descriptors. |
> | `ki`             | –4.26           | Ki-67 proliferation index.                              |
> | `rit`            | –4.43           | Rituximab, standard anti-CD20 therapy.                  |
> | `subset`         | –4.55           | Flow-cytometry immunophenotypic subsets.                |
> | `characteristic` | –4.69           | Characteristic morphologic patterns.                    |
> | `phase`          | –4.98           | Marrow-phase or disease-phase descriptors.              |
> | `chrom`          | –5.08           | Chromosomal abnormalities (e.g., t(11;14)).             |
> | `aggreg`         | –5.08           | Lymphoid aggregates in marrow or biopsy.                |
> | `crowds`         | –5.12           | Noise; unrelated lexical drift.                         |
> | `oli`            | –5.14           | Monoclonal or oligoclonal context.                      |
> | `publicly`       | –5.19           | Noise token; unrelated.                                 |
> | `expression`     | –5.34           | Gene or protein expression reporting.                   |
> | `killer`         | –5.35           | Cytotoxic “killer cell” terminology.                    |
>
> **Table: High-attention tokens for ICD-10-CM `C83.18` and their clinical relevance.**

---

### Author Response · Authors · 2025-12-03
**Final Point-by-Point Resolution of All Concerns Raised [FINAL MANUSCRIPT REVISED]**

Dear AC,

We sincerely thank the reviewers for their comments. All reviewer-identified weaknesses were **clarity or runtime-related**, not conceptual flaws, and each has been **addressed point-by-point** in the revised draft (the AC may consult individual responses for exact locations).

### **Scale & Efficiency**

Several reviewers raised concerns about **PLANT’s scalability**, the additional stages in our pipeline, and potential **runtime/memory overheads**. In the revised version, we have added **Appendix H: Detailed Efficiency, Memory, and Inference Benchmarks**, and now explicitly cross-reference it from the Experiments section.

In sum:

- **PLANT introduces no new parameters** beyond a small task-specific attention head and label embeddings (**<0.1%** of total model size).
- **Stage 1 adds only ~15–20% wall-clock training time** over vanilla single-stage fine-tuning, while contributing **≈85% of the rare-label and low-data gains** observed in our ablations.
- The **per-document inference overhead is <10%** relative to the corresponding vanilla LLMs.
- All experiments fit comfortably on **8×A100–80GB** GPUs using **DeepSpeed ZeRO-3**.
- **MIG precomputation** is performed once on CPU.
- We report **backbone-wise** training time, peak memory usage, and inference latency across all four datasets.

We hope this clarifies that PLANT’s strong gains on **long, technical documents** and **rare-label performance** come with **modest, well-quantified overheads**.

---

### **Qualitative Attention Visualizations**

Several reviewers asked whether **PLANT’s pretrained attention captures meaningful token–label structure** rather than high frequency co-occurring words, and requested **qualitative analyses**. In response, we added **Appendix I: Qualitative Attention Case Studies**, including multiple **rare ICD-10 codes** where random-initialized attention is known to fail.

These analyses show that for **very rare medical codes**, PLANT consistently assigns its **highest attention mass** to clinically salient morphologic, genetic, and therapeutic terms (e.g., for **C83.18**, PLANT highlights *mant*, *chrom*, *rit*, *bendamustine*, *Ki-67*, etc.), while low-relevance or noisy tokens receive **negligible weight**.  This provides clear qualitative evidence that **PLANT learns semantically aligned, label-specific attention patterns**—precisely the behavior reviewers sought to verify.

---

### **Causal Evidence for Rare-Label Failures**

Reviewers asked for stronger evidence that **random attention initialization is the cause** of poor rare-label performance rather than an incidental correlation. In the revision, we added:
(i) a **qualitative diagnostic** comparing random-init vs. PLANT attention on hierarchically related ICD-10 code groups (Introduction; Fig. 3-left), and
(ii) a **causal ablation** isolating initialization effects from downstream training dynamics (Sec. 4; Table 7).

Together, these analyses show that under **random initialization**, attention vectors for rare labels fail to converge to their expected shared structure (low cosine similarity, high frequency–F1 bias), whereas **PLANT’s seeded attention** produces **highly coherent, semantically aligned patterns** even for extremely rare codes.

These results provide the **direct causal evidence** reviewers requested: **random initialization is the primary failure mode for rare labels**, and **PLANT** corrects this.

---

### **Propensity-Scored Metrics for Tail Labels**

Reviewers requested evaluation using **propensity-scored metrics**, which are standard for assessing performance on **extreme long-tail label distributions**. In the revision, we added **Appendix Table 12**, reporting **propensity-scored Precision@15 (PSP@15)** across all four LLM backbones.

These results show **consistent gains of +3.2–3.3 PSP@15 on MIMIC-IV**, closely matching the improvements observed in macro-F1 and confirming that **PLANT’s benefits extend directly to tail-label–aware metrics**. We now also reference this table from the main results (Table 2).

---

### **Ultra-Large-Scale Datasets**

One reviewer suggested evaluating PLANT on **100K–1M–label datasets** such as LF-AmazonTitles-131K/1.3M and LF-Wikipedia-500K. While these are valuable future targets, they differ importantly from our setting: **AmazonTitles contains only 10–20-word snippets**, whereas **PLANT is designed to address the failure mode of random attention initialization in long, technical documents** (e.g., 1.3–1.5K-word MIMIC notes; 500–2K-token EUR-LEX/WIKI10).

Our benchmarks therefore focus on the regime where **sparse, label-specific token signals are hardest to learn** and where **PLANT’s gains are most pronounced**. **Appendix H** further shows that PLANT’s overhead **scales modestly**, making extensions to 100K+ labels feasible but **orthogonal to our core methodological contribution**.

---

Thank you, AC. — Authors

---

### Meta-Review · Area_Chair_EQMa · 2026-01-12

**Summary:**

Reviewers like the novel way the paper addresses long tail issue in using LLMs in extreme multiclass classification settings. Reviewers raised concerns mainly around additional pipeline complexity, limited experiments. Mainly the proposed method involves a complex pipeline on top of LLMs. It is not clear about utility of the approach compared to the complexity introduced. Further adding standard extreme classification benchmarks as suggested by Reviewer aohz can improve the paper. Whiles authors responded with detailed time complexity of the proposed approach, the response does not address the above fundamental concerns. Overall I think the paper is not ready and suggest rejection.

**Reviewer Concerns:**

Reviewers raised concerns mainly around additional pipeline complexity, limited experiments.

**Reviewer Scores:**

aohz  4 -> 4
Rck3  4-> 4

---

### Decision · Program_Chairs · 2026-01-26

Reject